# TOHAN: A One-step Approach towards Few-shot Hypothesis Adaptation

**Haoang Chi**[1,2,6]*, **Feng Liu**[3]*, **Wenjing Yang**[1]†, **Long Lan**[1,6]†, **Tongliang Liu**[4],
**Bo Han**[2], **William K. Cheung**[2], **James T. Kwok**[5]

[1] State Key Laboratory of High Performance Computing, College of CS, NUDT
[2] CS Department, HKBU
[3] DeSI Lab, AAII, Faculty of Engineering and IT, UTS
[4] TML Lab, School of CS, Faculty of Engineering, USYD
[5] CSE Department, HKUST
[6] Peng Cheng Laboratory, Shenzhen

`haoangchi618@gmail.com`, `feng.liu@uts.edu.au`, {`wenjing.yang`,
`long.lan`}`@nudt.edu.cn`, {`bhanml`, `william`}`@comp.hkbu.edu.hk`
`jamesk@cse.ust.hk`

## Abstract

In *few-shot domain adaptation* (FDA), classifiers for the target domain are trained with *accessible* labeled data in the *source domain* (SD) and few labeled data in the *target domain* (TD). However, data usually contain private information in the current era, e.g., data distributed on personal phones. Thus, the private data will be leaked if we directly access data in SD to train a target-domain classifier (required by FDA methods). In this paper, to prevent privacy leakage in SD, we consider a very challenging problem setting, where the classifier for the TD has to be trained using few labeled target data and a well-trained SD classifier, named *few-shot hypothesis adaptation* (FHA). In FHA, we cannot access data in SD, as a result, the private information in SD will be protected well. To this end, we propose a *target-oriented hypothesis adaptation network* (TOHAN) to solve the FHA problem, where we generate highly-compatible unlabeled data (i.e., an intermediate domain) to help train a target-domain classifier. TOHAN maintains two deep networks simultaneously, in which one focuses on learning an intermediate domain and the other takes care of the intermediate-to-target distributional adaptation and the target-risk minimization. Experimental results show that TOHAN outperforms competitive baselines significantly.

## 1 Introduction

In *domain adaptation* (DA) [7, 21, 41, 42, 50], we aim to train a target-domain classifier with data in source and target domains. Based on the availability of data in the target domain (e.g., fully-labeled data, partially-labeled data and unlabeled data), DA is divided into three categories: *supervised DA* (SDA) [43], semi-supervised DA [20] and *unsupervised DA* (UDA) [56]. Since SDA methods outperform UDA methods for the same quantity of target data [33], it becomes attractive if we can train a good target-domain classifier using labeled source data and few labeled target data [46].

Hence, *few-shot domain adaptation* (FDA) methods [33] are proposed to train a target-domain classifier with *accessible* labeled data from the source domain and few labeled data from the target

---

*Equal contribution. Work done when Haoang Chi remotely visited HKBU.
†Corresponding author.

35th Conference on Neural Information Processing Systems (NeurIPS 2021).

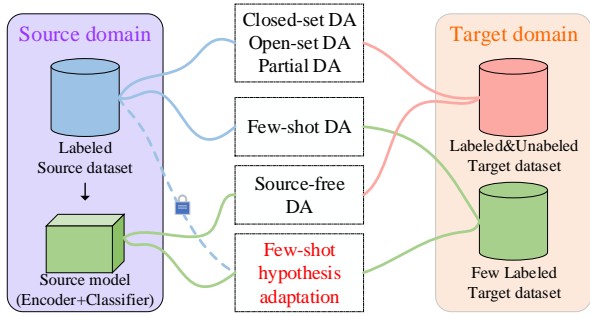

Figure 1: The *few-shot hypothesis adaptation* (FHA) and existing domain adaptation problem settings. In FHA, we aim to train a classifier for the target domain only using few labeled target data and a well-trained source-domain classifier. Namely, we do not access any source data when training the target-domain classifier. This setting prevents the data leakage of the source domain passively. The lock means we cannot access data in the source domain.

domain. Compared to SDA and UDA methods, FDA methods only require few data in the target domain, which is suitable to solve many problems, e.g., medical image processing [48]. Existing FDA methods involve many approaches and applications. Structural casual model [46] has been proposed to overcome the problem caused by apparent distribution discrepancy. Since deep neural networks tend to overfit the few-labeled data in the training process, a meta-learning method becomes an effective solution to the FDA problem [45]. Besides, FDA methods perform well in face generation [51] and virtual-to-real scene parsing [54].

However, it is risky to directly access source data for training a target-domain classifier (required by FDA methods) due to the private information contained in the source domain. In the current era, labeled data are distributed over different physical devices and usually contain private information, e.g., data on personal phones or from surveillance cameras [26]. Since FDA methods [46] require abundant labeled source data to train a target-domain classifier, they may leak private information in the training process, which may result in massive loss [19].

In this paper, to prevent the private data leakage of the source domain in existing FDA methods, we propose a novel and very challenging problem setting, where the classifier for the target domain has to be trained using few labeled target data and a well-trained source-domain classifier, named *few-shot hypothesis adaptation* (FHA, see Figure 1). In the literature [26], researchers have adapted a source-domain hypothesis to be a target-domain classifier when abundant unlabeled target data are available. However, since these methods require abundant target data, they cannot address the FHA problem well, which has been empirically verified in Table 1 and Table 2.

The key benefit of FHA is that we do not need to access the source data, which wisely avoids private-information leakage of source domain under mild assumptions. Besides, since the size of datasets of most domains is large in the real world, existing FDA methods will take a long time to train a target-domain classifier. However, in FHA, we train a target-domain classifier only with a source classifier and few labeled target data, reducing the computation cost greatly.

To address FHA, we first revisit the theory related to learning from few labeled data and try to find out if FHA can be addressed in principle. Fortunately, we find that, in *semi-supervised learning* (SSL) where only few labeled data available, researchers have already shown that, a good classifier can be learned if we have abundant unlabeled data that are compatible with the labeled data. Thus, motivated by the SSL, we aim to address FHA via gradually generating highly compatible data for the target domain. To this end, we propose a *target-oriented hypothesis adaptation network* (TOHAN) to solve the FHA problem. TOHAN maintains two deep networks simultaneously, in which one focuses on learning an intermediate domain (i.e., learning compatible data) and the other takes care of the intermediate-to-target distributional adaptation (Figure 2).

Specifically, due to the scarcity of target data, we cannot directly generate compatible data for the target domain. Thus, we first generate an intermediate domain where data are compatible with the given source classifier and the few labeled target data. Then, we conduct the intermediate-to-target

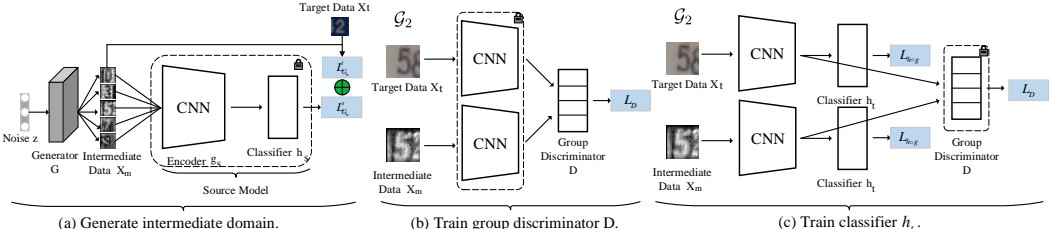

(a) Generate intermediate domain.  (b) Train group discriminator D.  (c) Train classifier $h_t$.

Figure 2: Overview of *target-oriented hypothesis adaptation network* (TOHAN). It consists of generator $G$, encoder $g_s$, $g_t$ (initialize $g_t=g_s$), classifier $h_s$, $h_t$ (initialize $h_t=h_s$) and group discriminator $D$. (a) Firstly, we train a generator $G$ using the source classifier $g_s$, $h_s$ and target data $D_t$. Then we generate intermediate data between the two domains. (b) We freeze $g_t$ and $h_t$ and update group discriminator $D$. (c) We freeze $D$ and update $g_t$ and $h_t$. In subfigures (b) and (c), they show a data pair from $\mathcal{G}_2$, where the two data points come from the same class but different domains.

distributional adaptation to make the generated intermediate domain close to the target domain. Eventually, we embed the above procedures into our one-step solution, TOHAN, to enable gradual generation of an intermediate domain that contains highly compatible data for the target domain. According to the learnability of SSL, with the generated "target-like" intermediate domain, TOHAN can learn a good target-domain classifier.

We conduct experiments on 8 FHA tasks on 5 datasets (*MNIST*, *SVHN*, *USPS*, *CIFAR*-10 and *STL*-10). We compare TOHAN with 5 competitive baselines. Experiments show that TOHAN effectively transfers knowledge of the source hypothesis to train a target-domain classifier when we only have few labeled target data. In other words, our paper opens a new door to the domain adaptation field, which solves private-data leakage and data shortage simultaneously.

## 2 Few-shot Hypothesis Adaptation

In this section, we formalize a novel and challenging problem setting, called *few-shot hypothesis adaptation* (FHA). Let $\mathcal{X} \subset \mathbb{R}^d$ be a feature (input) space and $\mathcal{Y} := \{1, \ldots, N\}$ be a label (output) space, and $N$ is the number of classes. A domain [9] for the FHA problem is defined as follows.

**Definition 1** (Domains for FHA). *Given random variables $X_s, X_t \in \mathcal{X}$, $Y_s, Y_t \in \mathcal{Y}$, the source and target domains are joint distributions $P(X_s, Y_s)$ and $P(X_t, Y_t)$, respectively, where the joint distributions $P(X_s, Y_s) \neq P(X_t, Y_t)$ and $\mathcal{X}$ is compact.*

Then the FHA problem is defined as follows.

**Problem 1** (FHA). *Given a model (consisting of an encoder $g_s$ and a classifier $h_s$) trained on the source domain $P(X_s, Y_s)$ and independent and identically distributed (i.i.d.) labeled data $D_t = \left\{ \left( x_t^i, y_t^i \right) \right\}_{i=1}^{n_t}$ ($n_t \leqslant 7N$, following [37]) drawn from the target domain $P(X_t, Y_t)$, the aim of FHA is to train a classifier $h_t : \mathcal{X} \to \mathcal{Y}$ with $g_s$, $h_s$ and $D_t$ such that $h_t$ can accurately classify target data drawn from $P(X_t, Y_t)$.*

**Remark 1.** In FHA, there exists an assumption: malicious attackers cannot easily find source-domain-like data from the Internet and via some other ways. Otherwise, attackers may use the attack methods [57] to recover the training data, leading to data leakage.

**Possible Privacy-leakage Issues in FHA.** The assumption in Remark 1 is derived from the attack methods that aim to recover training data from a well-trained model. According to recent model-inversion attack methods [57], they need to access auxiliary data whose background is similar to the training data to help recover input data. There also exists a white-box inference attack method [34] that determines a data point's membership in the training set of the model. Therefore, FHA belongs to passive protection, requiring the training data of source model are sufficiently different from public data. To thoroughly avoid this issue, data owners might utilize the defending techniques (against the model-inversion attacks) to train their source models.

**Comparison with Few-shot Learning.** The main difference between FHA and *few-shot learning* (FSL) is the representation of source domain. For FHA, source domain is represented by a model

trained with source data. While, for FSL, source domain is represented by labeled data themselves [29, 30]. Besides, the data used to train source classifiers come from different domains from target data in FHA, while source data and target data come from the same domain in FSL. The works [14, 49] propose to hallucinate additional training examples to solve few-shot visual recognition, inspired by human's visual imagination. Meta-learning [11, 40] also performs well in FSL by learning the distribution of tasks with high generalization ability. As using few data for training easily leads to overfitting, there are works [10, 55] trying to constrain the hypothesis space to avoid it. *Data augmentation generative adversarial network* (DAGAN) [2] aims to augment target data through a conditional generative adversarial network to enhance the few-shot learning procedure.

**Comparison with UDA.** The main differences between FHA and UDA lie in the amount and label of data in the two domains. For the source domain, UDA requires a large amount of labeled data [6, 59], while FHA only requires a well-trained model. For the target domain, UDA requires a large amount of unlabeled data [44, 60], while FHA requires *few* labeled data.

**Comparison with FDA.** With the development of FSL, researchers also apply ideas of FSL into domain adaptation, called *few-shot domain adaptation* (FDA). FADA [33] is a representative FDA method, which pairs data from the source domain and data from the target domain and then follows the adversarial domain adaptation method. Casual mechanism transfer [46] is another novel FDA method dealing with a meta-distributional scenario, in which the data generating mechanism is invariant among domains. Nevertheless, FDA methods still need to access many labeled source data for training, which may cause the private-information leakage of the source domain.

**Comparison with Hypothesis Transfer Learning.** In *hypothesis transfer learning* (HTL), we can only access a well-trained source-domain classifier and small labeled or abundant unlabeled target data. [24] requires small labeled target data and uses the Leave-One-Out error to find the optimal transfer parameters. Later, SHOT [26] is proposed to solve the HTL with many unlabeled target data by freezing the source-domain classifier and learning a target-specific feature extraction module. As for the universal setting, a two-stage learning process [23] has been proposed to address the HTL problem. Compared with FHA, HTL still requires at least small target data (e.g., at least 12 samples in binary classification problem [24], or at least $10\%$ target data are labeled [1]). In FHA, we focus on a more challenging situation: only few data (e.g., one sample per class) are available. Besides, previous solutions to HTL mainly focus on mortifying existing hypotheses or loss functions used for fine-tuning. However, our solution stems from the learnability of semi-supervised learning (Section 3) and try to generate more compatible data, which is quite different from previous works.

# 3   How to Learn from Few-shot Data in Principle

From the view of statistical learning theory [47], it is unrealistic to directly learn an accurate target-domain classifier only with few labeled data. However, the amount of labeled data in *semi-supervised learning* (SSL) [61] is also few (e.g., one sample per class), but SSL methods still achieves good performance across various learning tasks, which motivates us to consider solving FHA in the view of SSL. First, we will show theoretical analysis regarding learnability of SSL.

**Learnability of SSL.**   For simplicity, we consider the 0-1 semi-supervised classification problem. Let $c^* : \mathcal{X} \to \{0, 1\}$ be the optimal target classifier and $\mathcal{H} = \{h : \mathcal{X} \to \{0, 1\}\}$ is a hypothesis space. Let $err(h) = \mathbb{E}_{x \sim P}[h(x) \neq c^*(x)]$ be the true error rate of a hypothesis $h$ over a distribution $P$. In SSL, its learnability mainly depends on the compatibility $\chi : \mathcal{H} \times \mathcal{X} \mapsto [0, 1]$ that measures how "compatible" $h$ is to an unlabeled data $x$. Let $\chi(h, P) = \mathbb{E}_{x \sim P}[\chi(h, x)]$ be the expectation of compatibility of data from $P$ on a classifier $h$. If the unlabeled data and $c^*$ are highly compatible (i.e., $\chi(c^*, P)$ closes to 1), then, in theory, we can learn a good classifier with few labeled data and sufficient unlabeled data. Specifically, we have the following theorem (see proof in Appendix B).

**Theorem 1.** *Let* $\hat{\chi}(h, S) = \frac{1}{|S|} \sum_{x \in S} \chi(h, x)$ *be the empirical compatibility over unlabeled dataset* $S$. *Let* $\mathcal{H}_0 = \{h \in \mathcal{H} : \widehat{err}(h) = 0\}$. *If* $c^* \in \mathcal{H}$ *and* $\chi(c^*, P) = 1 - t$, *then* $m_u$ *unlabeled data and* $m_l$ *labeled data are sufficient to learn to error* $\epsilon$ *with probability* $1 - \delta$, *for*

$$m_u = \mathcal{O}\left( \frac{VCdim(\chi(\mathcal{H}))}{\epsilon^2} \log \frac{1}{\epsilon} + \frac{1}{\epsilon^2} \log \frac{2}{\delta} \right) \tag{1}$$

*and*

$$m_l = \frac{2}{\epsilon}\left[\ln(2\mathcal{H}_{P,\chi}(t+2\epsilon)[2m_l, P]) + \ln\frac{4}{\delta}\right], \tag{2}$$

*where $\chi(\mathcal{H}) = \{\chi_h : h \in \mathcal{H}\}$, $\chi_h(\cdot) = \chi(h, \cdot)$, and $\mathcal{H}_{P,\chi}(t+2\epsilon)[2m_l, P]$ is the expected number of splits of $2m_l$ data drawn from $P$ using hypotheses in $\mathcal{H}$ of compatibility more than $1 - t - 2\epsilon$. In particular, with probability at least $1 - \delta$, we have $err(\hat{h}) \leqslant \epsilon$, where*

$$\hat{h} = \underset{h \in \mathcal{H}_0}{\arg\max}\, \hat{\chi}(h, S). \tag{3}$$

**Remark 2.** If the unlabeled data are highly compatible to $c^*$, $t$ is small, which results in a smaller $m_l$. Namely, with the smaller $m_l$, we can still achieve a low error rate. In view of Theorem 1, it is clear that SSL will be learnable if many compatible unlabeled data are available. Motivated by SSL, we wonder if we can generate compatible data to help our learning task. The answer is *affirmative*.

**Solving FHA in Principle.** Motivated by Theorem 1, finding many highly compatible unlabeled data is a breakthrough point for FHA. Hence, generating unlabeled target data is a straightforward solution. However, due to the shortage of existing target data, directly generating them is unrealistic. To solve this problem, we can ask for help from the source classifier. In our paper, we first try to generate intermediate domain $P_m$ containing knowledge of source and target domains, which are compatible with both the source classifier and target classifier, i.e.,

$$P_m = \underset{P}{\arg\max}[\chi(h_s, P) + \chi(h_t, P)], \tag{4}$$

where $\chi(h_s, P)$ (resp. $\chi(h_t, P)$) measures how compatible $h_s$ (resp. $h_t$) is with the data distribution $P$. Then, we will adapt intermediate domain $P_m$ to the target domain via distributional adaptation with the training procedure going on. Finally, we can obtain many unlabeled data that are compatible with $h_s$ and $h_t$ (more compatible with $h_t$), meaning that, based on Theorem 1, we can address FHA in principle. According to Eq. (4), it can be seen that we can have two straightforward solutions: maximizing $\chi(h_s, P)$ or $\chi(h_t, P)$, corresponding to S+FADA and T+FADA in benchmark solutions. The results in Table 1 and Table 2 indicate that these two straightforward solutions cannot address FHA well, which motivates us to maximize them simultaneously, which is realized below.

## 4 Target-Oriented Hypothesis Adaptation Network for FHA Problem

This section presents a powerful one-step approach: *target-oriented hypothesis adaptation network* (TOHAN, see Figure 2). TOHAN can generate data that are highly compatible with both the source classifier and target classifier and adapt the knowledge of these data to the target domain gradually.

**Intermediate domain generation.** The first step of TOHAN is to generate the intermediate domain data (Figure 2a). We input Gaussian random noise $z$ to a generator $G_n$ (taking the $n^{th}$ class for an example), then the generator outputs generated data. We aim to generate data satisfying (1) the generated data $G_n(z)$ can be correctly classified by the given source classifier $f_s = h_s \circ g_s$, and (2) $G_n(z)$ becomes closer to the target domain with training procedure going on. Thus, there are two loss functions regarding the intermediate domain generation. The first one is as follows.

Without loss of the generality, we assume $G_n(z)$ generates $B$ images, where $B$ is the batchsize in the training process of TOHAN. When $G_n(z)$ is inputted to the source-domain classifier $f_s$, we will obtain an $B \times N$ matrix $\mathbf{G}_n^M$, where the $i^{th}$ row in $\mathbf{G}_n^M$ represents probability of the $i^{th}$ generated image belonging to each class. Thus, the $n^{th}$ column in $\mathbf{G}_n^M$ represents the probability that the $B$ generated images belongs to the $n^{th}$ class, and we denote the $n^{th}$ column in $\mathbf{G}_n^M$ as $l_n$. Since $G_n(z)$ aims to generate data belonging to the $n^{th}$ class, we should update parameters of $\mathbf{G}_n^M$ to make each element in $l_n$ close to 1. Namely, the first loss function to train the $G_n$ can be defined as

$$\mathcal{L}_{G_n}^s = \frac{1}{B}\|l_n - \mathbb{1}\|_2^2, \tag{5}$$

where $\mathbb{1}$ is a $B$-by-1 vector whose elements are 1.

As discussed before, we also want to reduce the distance between the generated data $G_n(z)$ and the target data whose labels are $n$. In this way, we can make the generated data close to the target

domain and attain an intermediate domain $P_m$. Following [27], we adopt an augmented $L_1$ distance $\|X - Y\|_1 = \sum_i \omega_i |X_i - Y_i|$, where $\omega_i = |X_i - Y_i|^2 / \|X - Y\|_2$. Compared to the ordinary $\ell_1$ norm, the augmented $L_1$ distance encourages larger gradients for feature dimensions with higher residual error [27]. Compared to the $\ell_2$ norm, since $L_1$ distance is more robust to outliers [36], it is better to measure the distance between generated images and target images. Thus, the second loss to train $G_n$ is defined as follows,

$$\mathcal{L}_{G_n}^t = \frac{1}{MBK} \sum_{i=1}^{B} \sum_{k=1}^{K} \left\| x_m^i - x_t^k \right\|_1, \tag{6}$$

where $M = \max_{x_1, x_2 \in \mathcal{X}} \|x_1 - x_2\|_1$ ($\mathcal{X}$ is compact and $\|\cdot\|_1$ is continuous) and $G_n(z) := \{x_m^i\}_{i=1}^{B}$. Combining Eq. (5) and Eq. (6), we obtain the total loss to train the generator $G_n$:

$$\mathcal{L}_{G_n} = \mathcal{L}_{G_n}^s + \lambda \mathcal{L}_{G_n}^t = \frac{1}{B} \|l_n - \mathbb{1}\|_2^2 + \frac{\lambda}{MBK} \sum_{i=1}^{B} \sum_{k=1}^{K} \left\| x_m^i - x_t^k \right\|_1, \tag{7}$$

where $\lambda$ is a hyper-parameter between two losses to tradeoff the weight of knowledge of source and target domains. To ensure that the generated data are high-quality images, we train the generator $G_n$ ($n = 1, \ldots, N$) for some steps all alone. Note that, Eq. (7) corresponds to Eq. (4), and Eq. (5), (resp. Eq. (6)) is corresponding to $\chi(h_s, P_m)$ (resp. $\chi(h_t, P_m)$). Then we conduct intermediate-to-target distributional adaptation (see the next paragraph) and generation simultaneously.

**Intermediate-to-target distributional adaptation.** Now, we focus on how to construct *domain-invariant representations* (DIP) between the intermediate domain and the target domain. Through DIP, a classifier for the intermediate domain can be used to classify target data well [28, 59].

Since we only have few target data per class, we aim to "augment" them. Following [33], we can overcome the shortage of target data by pairing them with the corresponding intermediate data. Specifically, we create 4 groups of data pairs: $\mathcal{G}_1$ consists of data pairs from the same domain with the same label, $\mathcal{G}_2$ consists of pairs from different domains (one from the intermediate and one from the target domain) but with the same label, $\mathcal{G}_3$ consists of pairs from the same domain with different labels, and $\mathcal{G}_4$ consists of pairs from different domains (one from the intermediate and one from the target domain) and with different labels.

Based on the above four groups, we construct a four-class group discriminator $D$ to decide which of the four groups a given data pair belongs to, which differs from classical adversarial domain adaptation [12, 20]. The group discriminator $D$ aims to classify the data pair groups. As a classification problem, we train $D$ with the standard categorical cross-entropy loss:

$$\mathcal{L}_D = -\hat{\mathbb{E}} \left[ \sum_{i=1}^{4} y_{\mathcal{G}_i} \log \left( D \left( \phi \left( \mathcal{G}_i \right) \right) \right) \right], \tag{8}$$

where $\hat{\mathbb{E}}[\cdot]$ represents the empirical mean value, $y_{\mathcal{G}_i}$ is the label of group $\mathcal{G}_i$, and $\phi(\mathcal{G}_i) := [g_t(x_1), g_t(x_2)]$, $(x_1, x_2) \in \mathcal{G}_i$, and $g_t$ is the encoder on target domain. Note that we freeze $g_t$ when minimizing the above loss function (see Figure 2b).

Next, we turn to train $g_t$ and $h_t$ with the group discriminator $D$ fixed, which confuses $D$ between $\mathcal{G}_1$ and $\mathcal{G}_2$ (also $\mathcal{G}_3$ and $\mathcal{G}_4$). However, we need $D$ to correctly discriminate positive pairs ($\mathcal{G}_1, \mathcal{G}_2$) from negative pairs ($\mathcal{G}_3, \mathcal{G}_4$). This means that domain confusion and classification are realized at the same time. We firstly initialize $g_t$ and $h_t$ with the same weight as $g_s$ and $h_s$, respectively. Motivated by the non-saturating game [13], we minimize the following loss to update $g_t$ and $h_t$ (see Figure 2c):

$$\mathcal{L}_{h \circ g} = -\beta \hat{\mathbb{E}} \left[ y_{\mathcal{G}_1} \log \left( D \left( \phi \left( \mathcal{G}_2 \right) \right) \right) - y_{\mathcal{G}_3} \log \left( D \left( \phi \left( \mathcal{G}_4 \right) \right) \right) \right] + \hat{\mathbb{E}} \left[ \ell \left( f_t \left( X_t \right), f_t^* (X_t) \right) \right], \tag{9}$$

where $\beta$ is a hyper-parameter to tradeoff confusion and classification and $\ell$ is the cross-entropy loss. $f_t := g_t \circ h_t$ is the target model and $f_t^*$ is the optimal target model. Corresponding to Theorem 1, optimizing the first term in Eq. (9) increases compatibility of the target model with the intermediate data, and optimizing the second term in Eq. (9) reduces $\widehat{err}(h_t)$, resulting in a smaller $err(h_t)$. Compared to [33], Eq. (9) means that we train the target model by confusing $D$ and improving classification accuracy simultaneously.

---

**Algorithm 1** Target-oriented hypothesis adaptation network (TOHAN)

---

**Input**: encoder $g_s$, classifier $h_s$, $D_t = \{x_t^i, y_t^i\}_{i=1}^{n_t}$, learning rate $\gamma_1$, $\gamma_2$, $\gamma_3$ and $\gamma_4$, total epoch $T_{max}$, pretraining $D$ epoch $T_d$, adaptation epoch $T_f$, network parameter $\{\theta_{G_n}\}_{n=1}^N$, $\theta_{h \circ g}$, $\theta_D$.

1: **Initialize** $\{\theta_{G_n}\}_{n=1}^N$ and $\theta_D$;
**for** $t = 1, 2, \ldots, T_{max}$ **do**
    2: **Initialize** $\mathcal{D}_m = \varnothing$
    **for** $n = 0, 1, \ldots, N - 1$ **do**
        3: **Generate** random noise $z$;
        4: **Generate** data $G_n(z)$ then **add** them to $\mathcal{D}_m$
        5: **Update** $\theta_{G_n} \leftarrow \theta_{G_n} - \gamma_1 \nabla \mathcal{L}_{G_n}(z, D_t)$ using Eq. (7);
    **end**
    **if** $t = T_{max} - T_f$ **then**
        **for** $i = 1, 2, \ldots, T_d$ **do**
            6: **Sample** $\mathcal{G}_1, \mathcal{G}_3$ from $\mathcal{D}_m \times \mathcal{D}_m$;
            7: **Sample** $\mathcal{G}_2, \mathcal{G}_4$ from $\mathcal{D}_m \times \mathcal{D}_t$;
            8: **Update** $\theta_D \leftarrow \theta_D - \gamma_2 \nabla \mathcal{L}_D\left(\{\mathcal{G}_i\}_{i=1}^4\right)$ using Eq. (8);
        **end**
    **end**
    **if** $t \geqslant T_{max} - T_f$ **then**
        9: **Sample** $\mathcal{G}_1, \mathcal{G}_3$ from $\mathcal{D}_m \times \mathcal{D}_m$;
        10: **Sample** $\mathcal{G}_2, \mathcal{G}_4$ from $\mathcal{D}_m \times \mathcal{D}_t$;
        11: **Update** $\theta_{h \circ g} \leftarrow \theta_{h \circ g} - \gamma_3 \mathcal{L}_{h \circ g}(\{\mathcal{G}_i\}_{i=1}^4, x_m, x_t)$ using Eq. (9);
        12: **Update** $\theta_D \leftarrow \theta_D - \gamma_4 \nabla \mathcal{L}_D\left(\{\mathcal{G}_i\}_{i=1}^4\right)$ using Eq. (8);
    **end**
**end**
**Output**: the neural network $h_t \circ g_t$.

---

**TOHAN: A one-step solution to FHA.** Although we can sequentially combine the above two steps to solve the FHA problem (i.e., a two-step solution), the fixed intermediate domain (generated by the first step) may have large distributional discrepancy with the target domain. As a result, such two-step solution may not obtain a good target-domain classifier. To address this issue, we introduce a one-step solution TOHAN. The ablation study verifies that TOHAN outperforms such two-step solution (see ST+F and TOHAN in Table 3).

The entire training procedure of TOHAN is shown in Algorithm 1. Since the convergence speed of generator $G$ is relatively slow, the quality of generated data is poor at the beginning of the training process of $G$. Thus, we will train the generator $G$ for a certain number of epochs before performing intermediate-to-target distributional adaptation (lines 2 to 5). When the generator can generate high-quality images, we train the generator and conduct adaptation together.

We train every generator $G_n$ ($n = 1, 2, \ldots, N$) separately, and we generate the intermediate domain data using the latest generators. Then, we pair the intermediate data with the target data and pretrain the group discriminator $D$ (lines 6 to 8). Next, we pair the intermediate data with target data and conduct the adaptation (lines 9 to 12). After conducting intermediate-to-target distributional adaptation, we obtain better $g_t$ and $h_t$, i.e. classifying the target data more accurately. With the better target-domain classifier, we can make the generated intermediate data get closer to the target domain, in turn, these generated intermediate data further promote adaptation performance.

## 5 Experiments

We compare TOHAN with benchmark solutions on five standard supervised DA datasets: *MNIST* (*M*), *SYHN* (*S*), *USPS* (*U*), *CIFAR*-10 (*CF*), *STL*-10 (*SL*). We follow the standard domain-adaptation protocols [39] and compare average accuracy of 5 independent repeated experiments. For digital datasets (i.e., *M*, *S*, and *U*), we choose the number of target data (per class) from 1 to 7 [33]. For objects datasets (i.e., *CF* and *SL*), we choose the number of target data as 10. Details regarding these datasets can be found in Appendix C. The code is available at github.com/Haoang97/TOHAN.

**Benchmark solutions for FHA.** Although the FHA is a new problem setting, we still design 5 benchmark solutions to this new problem. (1) *Without adaptation* (WA): to classify the target domain

Table 1: Classification accuracy±standard deviation (%) on 6 digits FHA tasks. Bold value represents the highest accuracy on each column.

| Tasks | WA | FHA Methods | Number of Target Data per Class | | | | | | |
|---|---|---|---|---|---|---|---|---|---|
| | | | 1 | 2 | 3 | 4 | 5 | 6 | 7 |
| $M{\rightarrow}S$ | 24.1 | FT | 26.7±1.0 | 26.8±2.1 | 26.8±1.6 | 27.0±0.7 | 27.3±1.2 | 27.5±0.8 | 28.3±1.5 |
| | | SHOT | 25.7±2.2 | 26.9±1.2 | 27.9±2.6 | 29.1±0.4 | 29.1±1.4 | 29.6±1.7 | 29.8±1.5 |
| | | S+F | 25.6±1.3 | 27.7±0.5 | 27.8±0.7 | 28.2±1.3 | 28.4±1.4 | 29.0±1.0 | 29.6±1.9 |
| | | T+F | 25.3±1.0 | 26.3±0.8 | 28.9±1.0 | 29.1±1.3 | 29.2±1.3 | 31.9±0.4 | 32.4±1.8 |
| | | TOHAN | **26.7±0.1** | **28.6±1.1** | **29.5±1.4** | **29.6±0.4** | **30.5±1.2** | **32.1±0.2** | **33.2±0.8** |
| $S{\rightarrow}M$ | 70.2 | FT | 70.2±0.0 | 70.6±0.3 | 70.7±0.1 | 70.8±0.3 | 70.9±0.2 | 71.1±0.3 | 71.1±0.4 |
| | | SHOT | 72.6±1.9 | 73.6±2.0 | 74.1±0.6 | 74.6±1.2 | 74.9±0.7 | 75.4±0.3 | 76.1±1.5 |
| | | S+F | 74.4±1.5 | 83.1±0.7 | 83.3±1.1 | 85.9±0.5 | 86.0±1.2 | 87.6±2.6 | 89.1±1.0 |
| | | T+F | 74.2±1.8 | 81.6±4.0 | 83.4±0.8 | 82.0±2.3 | 86.2±0.7 | 87.2±0.8 | 88.2±0.6 |
| | | TOHAN | **76.0±1.9** | **83.3±0.3** | **84.2±0.4** | **86.5±1.1** | **87.1±1.3** | **88.0±0.5** | **89.7±0.5** |
| $M{\rightarrow}U$ | 69.7 | FT | 74.4±0.7 | 76.7±1.9 | 76.9±2.2 | 77.3±1.1 | 77.6±1.4 | 78.3±2.1 | 78.3±1.6 |
| | | SHOT | 87.2±0.2 | 87.9±0.3 | 87.8±0.4 | 88.0±0.4 | 87.9±0.5 | 88.0±0.3 | 88.4±0.3 |
| | | S+F | 83.7±0.9 | 86.0±0.4 | 86.1±1.1 | 86.5±0.8 | 86.8±1.4 | 87.0±0.6 | 87.2±0.8 |
| | | T+F | 84.2±0.1 | 84.2±0.3 | 85.2±0.9 | 85.2±0.6 | 86.0±1.5 | 86.8±1.5 | 87.2±0.5 |
| | | TOHAN | **87.7±0.7** | **88.3±0.5** | **88.5±1.2** | **89.3±0.9** | **89.4±0.8** | **90.0±1.0** | **90.4±1.2** |
| $U{\rightarrow}M$ | 82.9 | FT | 83.5±0.4 | 84.3±2.4 | 84.5±0.7 | 85.5±1.3 | 86.6±1.0 | 87.2±0.7 | 88.1±2.7 |
| | | SHOT | 83.1±0.5 | **85.5±0.3** | **85.8±0.6** | 86.0±0.2 | 86.6±0.2 | 86.7±0.2 | 87.0±0.1 |
| | | S+F | 83.2±0.2 | 84.0±0.3 | 85.0±1.2 | 85.6±0.5 | 85.7±0.6 | 86.2±0.6 | 87.2±1.1 |
| | | T+F | 82.9±0.7 | 83.9±0.2 | 84.7±0.8 | 85.4±0.6 | 85.6±0.7 | 86.3±0.9 | 86.6±0.7 |
| | | TOHAN | **84.0±0.5** | 85.2±0.3 | 85.6±0.7 | **86.5±0.5** | **87.3±0.6** | **88.2±0.7** | **89.2±0.5** |
| $S{\rightarrow}U$ | 64.3 | FT | 64.9±1.1 | 66.5±1.5 | 66.7±1.7 | 67.3±1.1 | 68.1±2.3 | 68.3±0.5 | 69.7±1.4 |
| | | SHOT | 74.7±0.3 | 75.5±1.4 | 75.6±1.0 | 75.8±0.7 | 77.1±2.1 | 77.8±1.6 | 79.6±0.6 |
| | | S+F | 72.2±1.4 | 73.6±1.4 | 74.7±1.4 | 76.2±1.3 | 77.2±1.7 | 77.8±3.0 | 79.7±1.9 |
| | | T+F | 71.7±0.6 | 74.3±1.9 | 74.5±0.8 | 75.9±2.1 | 77.7±1.5 | 76.8±1.8 | 79.7±1.9 |
| | | TOHAN | **75.8±0.9** | **76.8±1.2** | **79.4±0.9** | **80.2±0.6** | **80.5±1.4** | **81.1±1.1** | **82.6±1.9** |
| $U{\rightarrow}S$ | 17.3 | FT | 23.4±1.8 | 23.6±2.7 | 23.8±1.6 | 24.6±1.4 | 24.6±1.2 | 24.8±0.7 | 25.5±1.8 |
| | | SHOT | **30.3±1.2** | **31.6±0.4** | 29.8±0.5 | 29.4±0.3 | 29.7±0.5 | 29.8±0.8 | 30.1±0.9 |
| | | S+F | 28.1±1.2 | 28.7±1.3 | 29.0±1.2 | 30.1±1.1 | 30.3±1.3 | 30.7±1.0 | 30.9±1.5 |
| | | T+F | 27.5±1.4 | 27.9±0.9 | 28.4±1.3 | 29.4±1.8 | 29.5±0.7 | 30.2±1.0 | 30.4±1.7 |
| | | TOHAN | 29.9±1.2 | 30.5±1.2 | **31.4±1.1** | **32.8±0.9** | **33.1±1.0** | **34.0±1.0** | **35.1±1.8** |

with the source classifier (encoder $g_s$ and classifier $h_s$). (2) *Fine-tuning* (FT): to train the *classifier $g_s$* with few owned target data. (3) *SHOT*: a novel HTL method, where we modify it to use the labeled target data instead of only using the unlabeled target data. [26]. (4) *S+FADA* (S+F): to generate faked source data with the source classifier then apply them to DANN [12]. (5) *T+FADA* (T+F): to generate fake target data with few real target data then apply them to DANN. We demonstrate details of 5 benchmark solutions in Appendix D. Experimental details can be found in Appendix E. Moreover, we conduct additional experiments to compare existing HTL method named dkdHTL [53], and the related results and analysis can be found in Appendix F.

**Results on digits FHA tasks.** We conduct experiments on 6 digits FHA tasks: $M{\rightarrow}S$, $S{\rightarrow}M$, $M{\rightarrow}U$, $U{\rightarrow}M$, $S{\rightarrow}U$ and $U{\rightarrow}S$. Table 1 reports the target-domain classification accuracy of 6 methods on 6 digits FHA tasks. It is clear that TOHAN performs the best on almost every task. On $M{\rightarrow}S$, $S{\rightarrow}M$, $M{\rightarrow}U$ and $S{\rightarrow}U$, TOHAN outperforms all benchmark solutions ob-

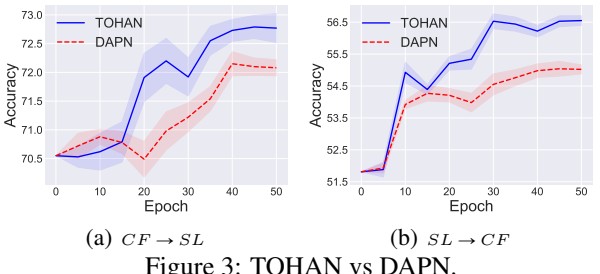

(a) $CF \rightarrow SL$    (b) $SL \rightarrow CF$

Figure 3: TOHAN vs DAPN.

viously. However, on the tasks $U{\rightarrow}M$ and $U{\rightarrow}S$, the accuracy of TOHAN is slightly lower than SHOT when the amount of target data is too small ($n = 1, 2$). This abnormal phenomenon shows that TOHAN cannot generate intermediate domain data effectively with very little target data, especially when the resolution of source data is much smaller than that of target data. In this case, the data we generate is close to the source domain, so TOHAN degrades to S+FADA.

Table 2: Classification accuracy±standard deviation (%) on 2 objects FHA tasks: CIFAR-10 → STL-10 (*CF→SL*) and STL-10 → CIFAR-10 (*SL→CF*). Bold value represents the highest accuracy (%) among TOHAN and benchmark solutions.

| Methods | WA | FT | SHOT | S+F | T+F | TOHAN |
|---------|-----|----|------|-----|-----|-------|
| *CF→SL* | 70.6 | 71.5±1.0 | 71.9±0.4 | 72.1±0.4 | 71.3±0.5 | **72.8±0.1** |
| *SL→CF* | 51.8 | 54.3±0.5 | 53.9±0.2 | **56.9±0.5** | 55.8±0.8 | 56.6±0.3 |

In Appendix G, we use t-SNE to visualize the features extracted by TOHAN and 5 benchmark solutions on $M→U$ task (see Figure 8 in Appendix G). When we use WA and FT methods, nearly all classes mix together. Although the classification accuracies of SHOT, S+F and T+F are relatively high, there are still some mixing among classes. For TOHAN, it can be seen that all classes are separated well, which demonstrates that TOHAN works well for solving the FHA problem.

**Results on objects FHA tasks.**   Following [39], we also evaluate TOHAN and benchmark solutions on 2 objects FHA tasks: $SL → CF$ and $CF → SL$, and the results are shown in Table 2. Considering the complexity of datasets and the difficulty of our problem setting, we do not have amazing results like digits tasks. In $SL → CF$, we achieve of $4.8\%$ improvement over WA and a performance accuracy of $56.9\%$. Note that because the numbers of pixels per image of $CF$ and $SL$ are quite different, the images from $SL$ lose a lot of information when inputted to the pre-trained model of $CF$, thus making the effects of TOHAN and benchmark solutions are not obvious for $CF → SL$.

**Comparing TOHAN with FSL methods.**   As mentioned above, FHA is a difficult case of FSL where the prior knowledge is a pre-trained model of another domain. To test the effectiveness of FSL methods in FHA, we compare TOHAN with a novel FSL method called *domain-adaptive few-shot learning* (DAPN) [58]. Note that we use the same pre-trained model in both TOHAN and DAPN. Taking $CF ↔ SL$ with five target data (per class) as an example, we solve FHA with TOHAN and DAPN and show the results in Figure 3. It is clear that TOHAN outperforms DAPN when the training epoch ($t$) is relatively large.

**Ablation Study.**   Finally, we study the advantages of one-step method over other two-step methods. We consider the following baselines: S+F, T+F and *ST+FADA* (ST+F). We have explained S+F and T+F previously. ST+F denotes the two-step version of TOHAN, i.e., to conduct intermediate domain generation and intermediate-to-target distributional adaptation separately. We make ablation study on three digital datasets mentioned before as an example.

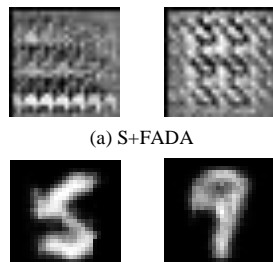

(a) S+FADA

(b) TOHAN

Figure 4: Visualization of S+FADA and TOHAN.

As shown in Table 3, it is clear that TOHAN works better than the other baselines. The generator of S+F uses the loss $\mathcal{L}_{G_n}^s$, which merely contains knowledge from the source domain. The generator of T+F uses the loss $\mathcal{L}_{G_n}^t$ and ignores the knowledge contained in the source-domain classifier. In contrast, TOHAN uses both $\mathcal{L}_{G_n}^s$ and $\mathcal{L}_{G_n}^t$. As a result, TOHAN achieves higher accuracy than S+F and T+F. Besides, the generators and classifiers in TOHAN promote each other in the training process, which results in that TOHAN performs better than the ST+F. In Figure 4, we visualize the data generated by S+FADA and TOHAN. It is clear that data generated by S+FADA are chaotic that contain little useful information. However, data generated by TOHAN contain many target-domain high-level visual features, and they can be classified by the source classifier accurately, resulting in a better performance in FHA. The detailed analysis of ablation study can be found in Appendix G.

**Verification of No Source-data Leakage in Intermediate Domain.**   As a key contribution, TOHAN solves FHA through generating intermediate data. To guarantee that no source data are leaked, we need to verify that there is no source-domain features in the intermediate data. We determine this by calculating the PSNR values [15, 52] between each intermediate sample and all source samples.

Table 3: Ablation study. We show the average accuracy of the 6 tasks on digits datasets in this table. Bold value represents the highest accuracy (%) on each column. See full results in Appendix G.

| FHA Methods | Number of Target Data per Class | | | | | | |
|---|---|---|---|---|---|---|---|
| | 1 | 2 | 3 | 4 | 5 | 6 | 7 |
| S+F | 61.2 | 63.0 | 64.3 | 65.4 | 65.7 | 66.4 | 67.2 |
| T+F | 61.0 | 63.0 | 64.2 | 64.5 | 65.7 | 66.5 | 67.4 |
| ST+F | 61.8 | 64.5 | 64.9 | 65.8 | 66.5 | 67.3 | 68.4 |
| TOHAN | **63.3** | **65.4** | **66.4** | **67.5** | **68.0** | **68.9** | **70.0** |

Table 4: The top-5 largest PSNR values between intermediate samples and all source samples.

| Ranking | 1 | 2 | 3 | 4 | 5 |
|---|---|---|---|---|---|
| PSNR value | 17.8951 | 17.8948 | 17.8948 | 17.8947 | 17.8947 |

PSNR indicates the generation quality of an image $f$ given a standard image $g$, and is defined as

$$\text{PSNR}(f,g) = 10\log_{10}\frac{255^2}{\text{MSE}(f,g)}, \text{ where } \text{MSE}(f,g) = \frac{1}{MN}\sum_{i=1}^{M}\sum_{j=1}^{N}(f_{ij}-g_{ij})^2.$$

The larger PSNR value indicates the two images are more similar. Therefore, taking $M \rightarrow S$ as an example, we report the top-5 largest PSNR values in Table 4. That is, we check whether the worst case satisfies our claim. For comparison, we also compute the PSNR values of (source data, source data), (target data, target data), (source data, target data), and (intermediate data, target data), and we report the *average* PSNR values of the above four cases in Figure 5.

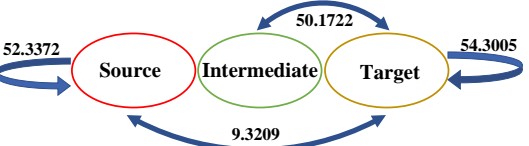

Figure 5: The average PSNR values of (source data, source data), (target data, target data), (source data, target data), and (intermediate data, target data).

As can be seen, the intermediate data are much closer to the target data, and they are very different from the source data. The average PSNR between source data and target data is 9.3209. The top-5 largest PSNR values between each intermediate data and all source data ($\approx 17.89$) are obviously smaller than 50.1722 (the average PSNR between intermediate data and target data). Through this result, we can state that intermediate data are similar to the target data and very different from the source data. Therefore, the above evidence shows that the generated intermediate data contain no source domain features, and the source data do not leak when generating the intermediate data.

# 6 Conclusion

This paper presents a very challenging problem setting called *few-shot hypothesis adaptation* (FHA), which trains a target-domain classifier with only few labeled target data and a well-trained source-domain classifier. Since we can only access a well-trained source-domain classifier in FHA, the private information in the source domain are protected well. To this end, we propose a novel one-step FHA method, called *target-oriented hypothesis adaptation network* (TOHAN). Experiments conducted on 8 FHA tasks confirm that TOHAN effectively adapts the source-domain classifier to the target domain and outperforms competitive benchmark solutions to the FHA problem.

**Acknowledgments and Disclosure of Funding**

This work was partially supported by the National Natural Science Foundation of China (No. 91948303-1, No. 61803375, No. 12002380, No. 62106278, No. 62101575, No. 61906210) and the National Grand R&D Plan (Grant No. 2020AAA0103501). FL would also like to thank Dr. Yanbin Liu for productive discussions.

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
