# A    Related Work

In this section, we briefly review *few-shot learning* (FSL) and two domain adaptation settings related to the FHA problem, which include FDA, and *source-data-free UDA* (SFUDA).

**Few-shot Learning.** Existing FSL methods can be divided into three categories: (1) Augmenting training data set by prior knowledge. Data augmentation via hand-crafted rules serves as pre-processing in FSL methods. For instance, we can use reflection [8]; and (2) Constraining hypothesis space by prior knowledge [32]; and (3) Altering search strategy in hypothesis space by prior knowledge. For instance, we can use early-stopping [3]. Note that our method belongs to category (1). However, the prior knowledge we have is more difficult to leverage than the prior knowledge that FSL methods have.

**Few-shot Domain Adaptation.** With the development of FSL, researchers also apply ideas of FSL into domain adaptation, called *few-shot domain adaptation* (FDA). FADA [33] is a representative FDA method, which pairs data from source domain and data from target domain and then follows the adversarial domain adaptation method. Casual mechanism transfer [46] is another novel FDA method dealing with a meta-distributional scenario, in which the data generating mechanism is invariant among domains. Nevertheless, FDA methods still need to access many labeled source data for training, which may cause the private-information leakage of the source domain.

**Hypothesis Transfer Learning.** In the *hypothesis transfer learning* (HTL), we can only access a well-trained source-domain classifier and small labeled or abundant unlabeled target data. [24] requires small labeled target data and uses the Leave-One-Out error find the optimal transfer parameters. Later, SHOT [26] is proposed to solve the HTL with many unlabeled target data by freezing the source-domain classifier and learning a target-specific feature extraction module. [16] proposes an image translation method that transfers the style of target images to that of unseen source images. As for the universal setting, a two-stage learning process [23] has been proposed to address the HTL problem. Compared with FHA, HTL still requires at least small target data (e.g., at least 12 samples in binary classification problem [24], or at least two of labeling percentage [1]). In FHA, we focus on a more challenging situation: only few data (e.g., one sample per class) are available.

# B    Proof of Theorem 1

We state here two known generalization bounds [5] used in our proof.

**Lemma 1.** *Suppose that $\mathcal{H}$ is a set of functions from $\mathcal{X}$ to $\{0, 1\}$ with finite $VC$-dimension $V \geqslant 1$. For any distribution $P$ over $\mathcal{X}$, any target function, and any $\epsilon$, $\delta > 0$, if we draw a set of data from $P$ of size*

$$m(\epsilon, \delta, V) = \frac{64}{\epsilon^2} \left( 2V \ln \left( \frac{12}{\epsilon} \right) + \ln \left( \frac{4}{\delta} \right) \right),$$

*then with probability at least $1 - \delta$, we have $|err(h) - \widehat{err}(h)| \leqslant \epsilon$ for all $h \in \mathcal{H}$.*

**Lemma 2.** *Suppose that $\mathcal{H}$ is a set of functions from $\mathcal{X}$ to $\{0, 1\}$ with finite $VC$-dimension $V \geqslant 1$. For any probability distribution $P$ over $\mathcal{X}$, any target function $c^*$, we have*

$$\boldsymbol{Pr} \left[ \sup_{h \in \mathcal{H}, \widehat{err}(h)=0} |err(h) - \widehat{err}(h) \geqslant \epsilon| \right] \leqslant 2\mathcal{H}[2m, P]e^{-m\epsilon/2}.$$

*So, for any $\epsilon$, $\delta > 0$, if we draw a set of data from $P$ of size*

$$m \geqslant \frac{2}{\epsilon} \left( 2 \ln(\mathcal{H}[2m, P]) + \ln \left( \frac{2}{\delta} \right) \right),$$

*then with probability at least $1 - \delta$, we have that all functions with $\widehat{err}(h) = 0$ satisfy*

$$err(h) \leqslant \epsilon.$$

Now we begin the proof of Theorem 1.

*Proof.* Let $S$ be the set of $m_u$ unlabeled data. By standard VC-dimension bounds (e.g., Lemma 1), the number of unlabeled data given is sufficient to ensure that with probability at least $1 - \frac{\delta}{2}$ we have

$$|\boldsymbol{Pr}_{x \sim \bar{S}}[\chi_h(x) = 1] - \boldsymbol{Pr}_{x \sim P}[\chi_h(x) = 1]| \leqslant \epsilon \quad \text{for all } \chi_h \in \chi(\mathcal{H}),$$

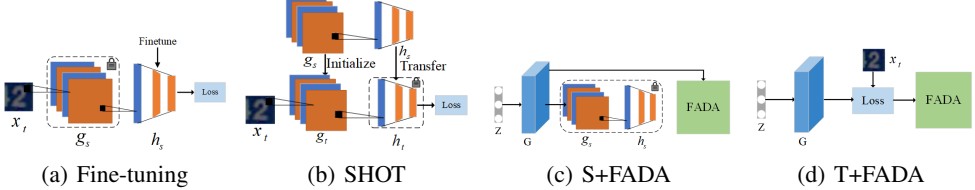

| (a) Fine-tuning | (b) SHOT | (c) S+FADA | (d) T+FADA |

Figure 6: Overview of benchmark solutions to the FHA problem. (a) We freeze the source encoder $g_s$ and train the source classifier $h_s$ with the target data $D_t$. (b) We first train source encoder $g_s$ and classifier $h_s$ , then we transfer them to the target domain. We generate pseudo labels for target data, then we use them to train the target model with classifier $h_t$ frozen. (c) We generate some source-like data under the guidance of the source classifier, then we combine them with FADA. (d) We generate some data close to target domain, i.e. decreasing the distance between data and target domain, then we combine them with FADA.

where $\bar{S}$ denotes the uniform distribution over $S$.

Since $\chi_h(x) = \chi(h, x)$, this implies that we have

$$|\chi(h, D) - \hat{\chi}(h, S)| \leqslant \epsilon \quad \text{for all } h \in \mathcal{H}.$$

Therefore, the set of hypotheses with $\hat{\chi}(h, S) \geqslant 1 - t - \epsilon$ is contained in $\mathcal{H}_{P,\chi}(t + 2\epsilon)$.

The bound on the number of labeled data now follows directly from known concentration results using the expected number of partitions instead of the maximum in the standard VC-dimension bounds (e.g., Lemma 2). This bound ensures that with probability $1 - \frac{\delta}{2}$, none of the functions $h \in \mathcal{H}_{P,\chi}(t + 2\epsilon)$ with $err(h) \geqslant \epsilon$ have $\widehat{err}(h) = 0$.

The above two arguments together imply that with probability $1 - \delta$, all $h \in \mathcal{H}$ with $\widehat{err}(h) = 0$ and $\hat{\chi}(h, S) \geqslant 1 - t - \epsilon$ have $err(h) \geqslant \epsilon$, and furthermore $c^*$ has $\hat{\chi}(c^*, S) \geqslant 1 - t - \epsilon$. This in turn implies that with probability at least $1 - \delta$, we have $err(\hat{h}) \leqslant \epsilon$, where

$$\hat{h} = \underset{h \in \mathcal{H}_0}{\arg\max} \, \hat{\chi}(h, S).$$

$\square$

## C  Datasets

**Digits.**  Following the evaluation protocol of [33], we conduct experiments on 6 adaptation scenarios: $M{\rightarrow}S$, $S{\rightarrow}M$, $M{\rightarrow}U$, $U{\rightarrow}M$, $S{\rightarrow}U$ and $U{\rightarrow}S$. MNIST [25] images have been size-normalized and centered in a fixed-size ($28 \times 28$) image. USPS [18] images are $16 \times 16$ grayscale pixels. SVHN [35] images are $32 \times 32$ pixels with 3 channels.

**Objects.**  We also evaluate TOHAN and benchmark solutions on CIFAR-10 [22] and STL-10 [4], following [39]. The CIFAR-10 dataset contains $60,000$ $32 \times 32$ color images in 10 categories. The STL-10 dataset is inspired by the CIFAR-10 dataset but with some modifications. However, these two datasets only contain nine overlapping classes. We removed the non-overlapping classes ("frog" and "monkey") [39].

## D  Benchmark Solutions for FHA

To solve the FHA problem, this section presents 5 benchmark solutions that directly combine existing techniques used in the deep learning and domain adaptation fields.

**Without adaptation.**  Since we have a source-domain classifier, we can directly use it to classify the target data, which is a frustrating solution to the FHA problem.

**Fine-tuning.**  See Figure 6(a). Fine-tuning is a basic solution to the FHA problem. We freeze the source encoder $g_s$ and train the source classifier $h_s$ with the target data $D_t$. In this way, knowledge about target domain is filled into source hypothesis.

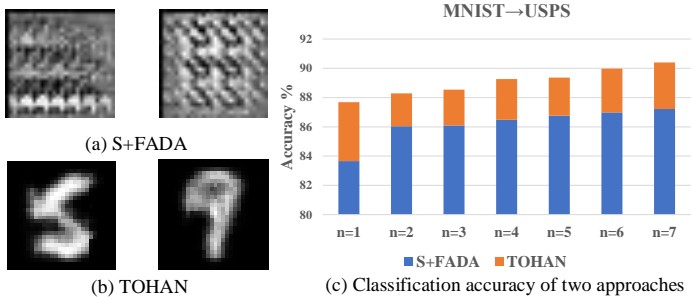

(a) S+FADA

(b) TOHAN

(c) Classification accuracy of two approaches

Figure 7: A straightforward solution for FHA is a two-step approach. Namely, we can first generate source data and then train a target-domain classifier using the generated source data and an FDA method (e.g., *few-shot adversarial domain adaptation* (FADA)). A visualized comparison between S+FADA and TOHAN (take MNIST→USPS as an example) is displayed in subfigures (a) and (b). On the left side, Subfigure (a) illustrates source-domain data generated by a two-step method: S+FADA. It is clear that the generated data are just noise and do not contain useful information about the source domain. In subfigure (b), we illustrate the intermediate-domain data generated by our method (i.e., TOHAN). It is clear that the generated intermediate-domain data contain useful information about two domains. In subfigure (c), the histogram shows classification accuracy of the two methods, and TOHAN outperforms S+FADA clearly.

**SHOT.** See Figure 6(b). SHOT is a novel method for source hypothesis transfer [26]. It learns the optimal target-specific feature learning module to fit the source hypothesis with only the source classifier. We first train source encoder $g_s$ and classifier $h_s$, and then we transfer them to the target domain. SHOT is an UDA method. Thus, we generate pseudo labels for target data, and then we use them to train the target model with classifier $h_t$ freezed. Although SHOT is suitable for our FHA problem, it requires a lot of target data, which is an obstacle for FHA.

**S+FADA.** See Figure 6(c). As mentioned in Figure 7, a straightforward solution to the FHA problem is a two-step approach. We can train a source-data generator $G$ under the guidance of source hypothesis, and then we use it to generate source data. First, we input Gaussian random noise $z$ to $G$, then $G$ outputs various disordered data. Second, these data is inputted into $g_s \circ h_s$, and then $h_s$ outputs the probability of $G(z)$ belonging to each class. Third, if we would like to generate data belonging to $n^{th}$ class, we should optimize $G$ to push the probability of $G(z)$ belonging to $n$ near to 1. Finally, we can apply the restored source data into an adversarial DA method to train a target domain classifier $h_t$.

**T+FADA.** See Figure 6(d). Different from S+FADA, we train a generator with the help of target data instead, and then we generate data close to target domain. We input Gaussian random noise $z$ to generator $G$ and minimize the distance between $G(z)$ and target data. Finally, we sequentially combine these generated data with adversarial DA method to train a target-domain classifier.

## E    Implementation Details

We implement all methods by PyTorch 1.7.1 and Python 3.7.6, and conduct all the experiments on two NVIDIA RTX 2080Ti GPUs.

**Network architecture.** We select architecture of generators $G_n$ ($n = 1, \ldots, N$) from DCGAN [38]. For digits tasks, the encoder $g$, classifier $h$ and group discriminator $D$ share the same architecture in all 6 tasks, following FADA [33]. As for encoder $g$, we employ the backbone network of LeNet-5 with batch normalization and dropout. For classifier $h$, we adopt one fully connected layer with softmax fuction. For group discriminator $D$, we adopt 3 connected layers with softmax function. For objects tasks, we employ Densenet-169 [17] as encoder $g$.

**Network hyper-parameters.** We set fixed hyper-parameters in every method which is irrelevant to dataset, based on the common protocol of domain adaptation [39]. The batch size of generator

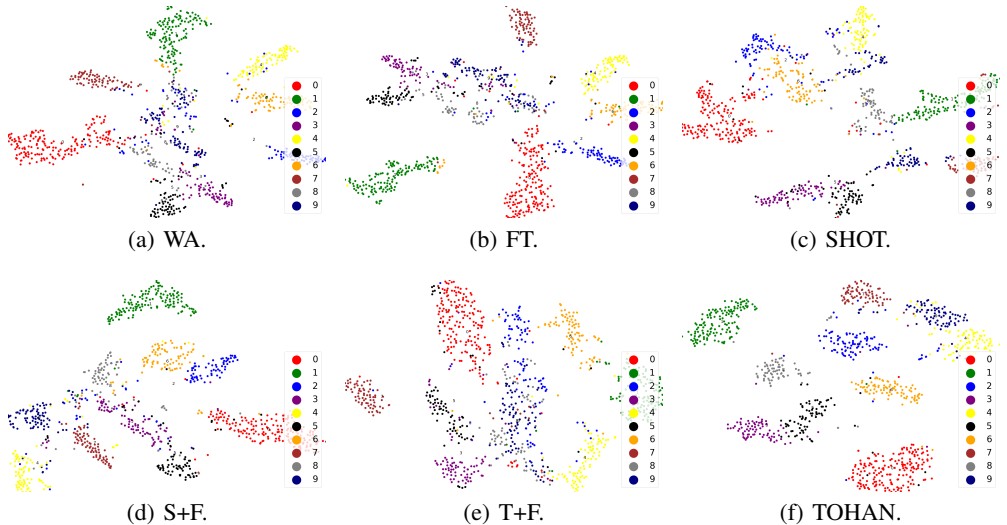

Figure 8: The t-SNE visualization for a 10-way classification task (taking *MNIST→USPS* as an example). When we use WA and FT methods, nearly all classes mix together. Although the classification accuracy of SHOT, S+F and T+F are relatively high, there are still a little mixtures among classes. For TOHAN, it can be seen that all classes are separated well. Namely, TOHAN works well for solving FHA problem.

is set to 32, and the batch size of group discriminator, encoder, classifier is all 64. We pre-train the group discriminator for 100 epochs. Meanwhile, the numbers of training steps of generator, group discriminator, encoder, classifier are set to 500, 50, 50, 50, respectively. Adam optimizer is with the same learning rate of $1 \times 10^{-3}$ in generators, encoder, classifier and group discriminator. The tradeoff parameter $\beta$ in Eq. (9) is set to $\frac{2}{1+\exp(-10\hat{q})} - 1$, same as [31]. And the tradeoff parameter $\lambda$ in Eq. (7) is set to 0.2 fixed. For the fair comparisons, we only resize and normalize the image and do not use any addition data augment or transformation. Note that, for each experiment, we report the result of the model trained in the last epoch.

## F   Additional Experiments about HTL

In this section, we compare TOHAN with another novel HTL method, i.e., *dynamic knowledge distillation for HTL* (dkdHTL) [53]. It is worth noting that dkdHTL is a black-box HTL method. That is, we cannot access the parameters of source model. Therefore, in distillation loss, we cannot get the logits, which is used to compute the soften probabilities with a high temperature $T > 1$. To address this problem, they tried to solve the logits through soften probabilities approximately. For the sake of fairness, we convert dkdHTL to white-box version. Specifically, we use the standard softmax function in distillation loss, instead of the approximate version. Moreover, we initial the parameters of target model by source model. Then, we show the results of dkdHTL in Table 5 and Table 6.

We find that TOHAN outperforms dkdHTL in most tasks significantly. However, in $S{\rightarrow}M$, $U{\rightarrow}M$, and $S{\rightarrow}U$, there exists few subtasks that dkdHTL outperforms TOHAN. In these three tasks, the complexity of source domain is high, while the complexity of target domain is low. TOHAN cannot generate qualified intermediate data effectively when the number of target data is very few and the source domain is highly complex simultaneously. dkdHTL is only suitable for tasks with uncomplicated target domain. The main reason is that the training data of dkdHTL are only the few target data. If the target domain is complex, it is very easy to overfit. Therefore, as for tasks with complex target domains, TOHAN has the upper hand.

Table 5: Comparison between dkdHTL and TOHAN. We report the classification accuracy±standard deviation (%) on 6 digits FHA tasks. Bold value represents the highest accuracy on each column.

| Tasks | FHA Methods | Number of Target Data per Class | | | | | | |
|---|---|---|---|---|---|---|---|---|
| | | 1 | 2 | 3 | 4 | 5 | 6 | 7 |
| $M{\rightarrow}S$ | dkdHTL | 24.1±0.7 | 24.1±0.3 | 24.5±0.6 | 24.4±1.1 | 25.4±0.8 | 25.7±0.5 | 26.1±1.1 |
| | TOHAN | **26.7±0.1** | **28.6±1.1** | **29.5±1.4** | **29.6±0.4** | **30.5±1.2** | **32.1±0.2** | **33.2±0.8** |
| $S{\rightarrow}M$ | dkdHTL | 71.2±1.2 | **83.4±0.4** | **88.5±0.6** | **88.2±0.7** | **89.5±0.7** | **89.6±0.4** | **90.3±0.2** |
| | TOHAN | **76.0±1.9** | 83.3±0.3 | 84.2±0.4 | 86.5±1.1 | 87.1±1.3 | 88.0±0.5 | 89.7±0.5 |
| $M{\rightarrow}U$ | dkdHTL | 65.2±0.6 | 70.5±1.3 | 74.4±0.6 | 77.8±0.6 | 78.6±0.9 | 78.8±1.1 | 79.0±1.3 |
| | TOHAN | **87.7±0.7** | **88.3±0.5** | **88.5±1.2** | **89.3±0.9** | **89.4±0.8** | **90.0±1.0** | **90.4±1.2** |
| $U{\rightarrow}M$ | dkdHTL | 83.2±0.2 | **85.5±0.5** | **85.9±0.4** | 85.7±0.8 | 86.2±0.2 | 86.2±0.4 | 86.8±0.3 |
| | TOHAN | **84.0±0.5** | 85.2±0.3 | 85.6±0.7 | **86.5±0.5** | **87.3±0.6** | **88.2±0.7** | **89.2±0.5** |
| $S{\rightarrow}U$ | dkdHTL | **76.3±0.3** | **77.6±0.5** | 78.9±0.4 | 79.5±0.4 | 80.2±0.5 | 80.7±0.4 | 82.1±0.4 |
| | TOHAN | 75.8±0.9 | 76.8±1.2 | **79.4±0.9** | **80.2±0.6** | **80.5±1.4** | **81.1±1.1** | **82.6±1.9** |
| $U{\rightarrow}S$ | dkdHTL | 20.5±0.8 | 20.9±0.4 | 21.7±0.3 | 23.8±0.2 | 24.5±0.7 | 25.5±0.6 | 25.7±0.4 |
| | TOHAN | **29.9±1.2** | **30.5±1.2** | **31.4±1.1** | **32.8±0.9** | **33.1±1.0** | **34.0±1.0** | **35.1±1.8** |

Table 6: Comparison between dkdHTL and TOHAN. We report the classification accuracy±standard deviation (%) on 2 objects FHA tasks: CIFAR-10 → STL-10 and STL-10 → CIFAR-10. Bold value represents the highest accuracy (%) among TOHAN and benchmark solutions.

| Tasks | dkdHTL | TOHAN |
|---|---|---|
| CIFAR-10 → STL-10 | 70.8±0.7 | **72.8±0.1** |
| STL-10 → CIFAR-10 | 52.4±0.5 | **56.6±0.3** |

# G  Additional Analysis

**Visualization of Results.**   We use t-SNE to visualize the feature (the penultimate layer of the classifier) extracted by TOHAN and 5 benchmark solutions on $M{\rightarrow}U$ task (see Figure 8). When we use WA and FT methods, nearly all classes mix together. Although the classification accuracy of SHOT, S+F and T+F are relatively high, there are still a little mixtures among classes. For TOHAN, all classes are separated well, which demonstrates that TOHAN works well for solving FHA problem.

**Detailed Analysis of Ablation Study.**   Table 7 shows the full results of ablation study. It is clear that TOHAN performs better than the corresponding two-step approach ST+FADA. However, when the number of target data is too small, ST+FADA may outperform TOHAN with a small probability. The reason for this abnormal phenomenon may be the limitation of target data. Although we use the technique of paring data, overfitting still occurs when data are scarce.

# H  Limitations

The main limitation in this paper is that the run time of TOHAN is a little long. The main reason causing the long run time is the generation part of TOHAN. Specifically, the second term of Eq. (7) is time-consuming, as we need to calculate the distances between each intermediate data and each target data. We will optimize the generation part to overcome the time-consuming problem.

Although the generation part of TOHAN is a little time consuming, it solves the challenge of lacking source data in FHA efficiently. TOHAN can generate intermediate data containing the knowledge of source domain and target domain. Therefore, we not only adapt more useful source domain knowledge to target domain, but also prevent the privacy leakage of source domain.

# I  Potential Negative Societal Impacts

The main potential negative societal impact in this paper is that TOHAN has a certain randomness. This is, TOHAN may not perform well consistently across various tasks. For example, TOHAN may

Table 7: Ablation Study. Bold value represents the highest accuracy (%) on each column. Data behind '±' is the standard derivation.

| Tasks | FHA Methods | Number of target data | | | | | | |
|---|---|---|---|---|---|---|---|---|
| | | 1 | 2 | 3 | 4 | 5 | 6 | 7 |
| $M{\to}S$ | S+FADA | 25.6±1.3 | 27.7±0.5 | 27.8±0.7 | 28.2±1.3 | 28.4±1.4 | 29.0±1.0 | 29.6±1.9 |
| | T+FADA | 25.3±1.0 | 26.3±0.8 | 28.9±1.0 | 29.1±1.3 | 29.2±1.3 | 31.9±0.4 | 32.4±1.8 |
| | ST+FADA | 25.7±0.7 | 28.1±0.9 | 28.5±1.2 | 29.2±1.0 | 29.2±0.8 | 31.3±1.7 | 32.0±0.8 |
| | TOHAN | **26.7±0.1** | **28.6±1.1** | **29.5±1.4** | **29.6±0.4** | **30.5±1.2** | **32.1±0.23** | **33.2±0.8** |
| $S{\to}M$ | S+FADA | 74.4±1.5 | 83.1±0.7 | 83.3±1.1 | 85.9±0.5 | 86.0±1.2 | 87.6±2.6 | 89.1±1.0 |
| | T+FADA | 74.2±1.8 | 81.6±4.0 | 83.4±0.8 | 82.0±2.3 | 86.2±0.7 | 87.2±0.8 | 88.2±0.6 |
| | ST+FADA | 74.3±1.2 | **83.7±1.0** | 83.8±0.8 | 85.8±0.6 | 86.0±0.9 | 87.7±0.8 | 89.0±0.6 |
| | TOHAN | **76.0±1.9** | 83.3±0.3 | **84.2±0.4** | **86.5±1.1** | **87.1±1.3** | **88.0±0.5** | **89.7±0.5** |
| $M{\to}U$ | S+FADA | 83.7±0.9 | 86.0±0.4 | 86.1±1.1 | 86.5±0.8 | 86.8±1.4 | 87.0±0.6 | 87.2±0.8 |
| | T+FADA | 84.2±0.1 | 84.2±0.3 | 85.2±0.9 | 85.2±0.6 | 86.0±1.5 | 86.8±1.5 | 87.2±0.5 |
| | ST+FADA | 86.1±1.5 | 87.1±1.6 | 86.9±0.7 | 87.9±1.1 | 88.0±1.2 | 88.3±0.7 | 88.5±1.3 |
| | TOHAN | **87.7±0.7** | **88.3±0.5** | **88.5±1.2** | **89.3±0.9** | **89.4±0.8** | **90.0±1.0** | **90.4±1.2** |
| $U{\to}M$ | S+FADA | 83.2±0.2 | 83.9±0.3 | 84.9±1.2 | 85.6±0.5 | 85.7±0.6 | 86.2±0.6 | 87.2±1.1 |
| | T+FADA | 82.9±0.7 | 83.9±0.2 | 84.7±0.8 | 85.4±0.6 | 85.6±0.7 | 86.3±0.9 | 86.6±0.7 |
| | ST+FADA | **84.0±0.7** | 84.2±0.5 | 85.3±1.0 | 85.6±1.2 | 86.7±1.0 | 86.5±0.5 | 88.0±1.0 |
| | TOHAN | 84.0±0.5 | **85.2±0.3** | **85.6±0.7** | **86.5±0.5** | **87.3±0.6** | **88.2±0.7** | **89.2±0.5** |
| $S{\to}U$ | S+FADA | 72.2±1.4 | 73.6±1.4 | 74.7±1.4 | 76.2±1.3 | 77.2±1.7 | 77.8±3.0 | 79.7±1.9 |
| | T+FADA | 71.7±0.6 | 74.3±1.9 | 74.5±0.8 | 75.9±2.1 | 77.7±1.5 | 76.8±1.8 | 79.7±1.9 |
| | ST+FADA | 73.1±0.9 | 75.2±1.3 | 75.9±0.8 | 76.3±1.5 | 78.3±1.6 | 79.1±1.7 | 79.7±1.6 |
| | TOHAN | **75.8±0.9** | **76.8±1.2** | **79.4±0.9** | **80.2±0.6** | **80.5±1.4** | **81.1±1.1** | **82.6±1.9** |
| $U{\to}S$ | S+FADA | 28.1±1.2 | 28.7±1.3 | 29.0±1.2 | 30.1±1.1 | 30.3±1.3 | 30.7±1.0 | 30.9±1.5 |
| | T+FADA | 27.5±1.4 | 27.9±0.9 | 28.4±1.3 | 29.4±1.8 | 29.5±0.7 | 30.2±1.0 | 30.4±1.7 |
| | ST+FADA | 28.1±1.3 | 28.9±0.7 | 29.2±1.5 | 29.8±1.2 | 31.0±0.9 | 31.2±0.9 | 33.2±1.7 |
| | TOHAN | **29.9±1.2** | **30.5±1.2** | **31.4±1.1** | **32.8±0.9** | **33.1±1.0** | **34.0±1.0** | **35.1±1.8** |

fail to adapt knowledge between two domains that have a large discrepancy. Therefore, if TOHAN makes a mistake in a critical area, the consequences will be bad.