# OpenReview forum: "TOHAN: A One-step Approach towards Few-shot Hypothesis Adaptation"
_NeurIPS.cc/2021/Conference — NeurIPS 2021 Spotlight_

### Official Review · Reviewer_1Jpd · 2021-07-12

**Rating:** 7
**Confidence:** 4

**Summary:**

This paper proposes a novel problem in domain adaptation - Few-shot Hypothesis Adaptation (FHA) - that combines Hypothesis Transfer Learning and the Few-Shot Domain Adaptation problem. Specifically, it assumes a lack of source domain data but the presence of a classifier (hypothesis) pretrained on the source domain and a few labelled data points (<=10 per class) from the target domain (with no additional data). The paper also adapts the FADA method and proposes a novel method, called TOHAN, which uses the source-domain hypothesis and the few target-domain data points to generate the intermediate-domain data compatible with the two domains.

**Limitations And Societal Impact:**

The authors claim that their method allows for full privacy of the source domain by relying on a well-trained source domain classifier. However, this seems to be based on the false assumption that the source domain classifier does not leak any private information. Recent studies (for example, Nasr et al. (2019) and Ateniese et al. (2015)) suggest that there might be a significant amount of source-domain information that pre-trained models leak. Moreover, TOHAN relies on leaking information from the source domain to generate realistic/compatible intermediate-domain data. This completely invalidates one of the main claims of the paper that private information is protected.

**Main Review:**

\+ means strength, \- means weakness

**Originality**

\+ *Novel setting*. As far as I am aware, the paper proposes a novel setting - Few-shot Hypothesis Adaptation (FHA) - a combination of existing problems - Hypothesis Transfer Learning and the Few-Shot Domain Adaptation.

\+/\- *Somewhat novel method*. As far as I am aware, the paper also proposes a novel method - TOHAN - which is a minor adaptation of FADA [23] into the setting. The method architecture seems to be heavily inspired by FADA [23]. Unlike FADA, TOHAN generates and uses the generated intermediate domain instead of the original source domain. However, apart from that, it is not entirely clear what the technical differences are between these methods. Which line in Algorithm 1 would be different for FADA / S-FADA / T-FADA / ST-FADA?

\- *The relation between FHA and FSL is poorly explained, and FSL is poorly cited.* I found the sentence in lines 88-89 particularly vague and poorly written. In addition, line 90 that compares TSN [37] and ProtoNets [29] ("which is relatively weaker than the former") brings little value to the paper. In my humble opinion, these methods were designed for two different problems (i.e. video action classification [37] and single image classification [29]), and it is inappropriate to compare them in this way. The authors might find the following works from FSL literature more related to their work: (Antreas et al., 2018), (Hariharan & Girshick., 2017), (Wang et al., 2018). Generally, these methods also generate/hallucinate samples from limited target-domain data and should be cited.

**Quality**

\+ *Some good experiments.* The paper performs a solid number of comparisons with representative baselines from HTL and FDA literature ([19] and [23], respectively) and basic fine-tuning baseline proposed by the authors.

\+/\- *Work seems grounded in some theoretical work*. However, I did not attempt to verify the proof, so I cannot comment on its correctness.

\- *Only marginal improvements over baselines, mostly within the error bar range.* Although the authors claim the method performs better than the baselines, the error range is rather high, suggesting that the performance differences between some methods are not very significant.

\- *There might be a possible fundamental flaw to the claim of full data privacy.* The authors claim that the TOHAN method (and therefore also the FHA problem) "strictly" protects the privacy of the source domain by using the source hypothesis rather than the source data itself (lines 245-247). However, the claim that knowledge is "completely" inaccessible may be false. For instance, Ateniese et al. (2015) have shown that it is possible to extract certain types of information from pretrained classifiers and models. In a more recent privacy analysis of deep learning, Nasr et al. (2019) suggest that even well-trained models might leak a significant amount of information about the data they were trained on. Moreover, the proposed TOHAN relies on the leaked source-domain knowledge to generator appropriate source-domain data. Therefore, it seems to me that claim that TOHAN is privacy secure is completely false.

\- *Lack of sufficient evidence for no source domain features in intermediate data.* The authors claim that "high-level, visual and useful features of source domain are rare in the generated intermediate data (Figure 6)" (lines 243-244); however, no empirical value is provided to support this claim (i.e. what do authors mean by "rare" in this context? how rare is "rare"?). Figure 6a/b does support this claim as it shows only 4 intermediate-domain images and no examples of the original source-domain data. This is insufficient evidence to draw the conclusion in lines 243-244, and in fact, it points to the contrary.

**Clarity**

\- *The paper is poorly written.* The paper is not particularly easy to read. It contains many spelling and grammatical errors (see below for proposed corrections). There is a large number of acronyms (e.g. FHA, SHOT, HTL, ST+F etc..) and some confusing mathematical notation (e.g. $D$, $\mathcal{D}$, $X$ and $\mathcal{X}$ refer to different entities) which make this work more confusing to read. The notation in the figure is inconsistent with the main text (uses $\mathcal{X}$ instead of $\mathcal{D}$).

**Significance**

\- *The paper presents a novel and interesting problem but could be flawed.* This novel problem and method could have important consequences in the context of data privacy - however, to me, the idea seems fundamentally flawed (see my comments in the "Quality" subsection, or  "Limitations And Societal Impact").

\- *The TOHAN improvements over baselines are mostly marginal.* Although the authors claim the method performs better than the baselines, the error range is rather high, suggesting that the performance differences between some methods are not very significant and the improvements are marginal.


**Spelling, Grammer, and Other Possible Mistakes.**

\- Line 32, grammar: "face of generation" --> "face generation"

\- Figure 2 caption, grammar: "which two data come from ... " --> "where the two data points come from ... "

\- Line 128, wrong word: "dependents" --> "depends"

\- Line 167, redundant word: "regarding to the ..." --> "regarding the ... "

\- Line 210, redundant words: "which confuses D unable to distinguish between ..." --> "which confuses D between ... "

\- Line 213, wrong word: "initial" --> "initialize"

**References**

Ateniese et al., 2015, "Hacking smart machines with smarter ones: How to extract meaningful data from machine learning classifiers", International Journal of Security and Networks, Volume 10, Issue 3

Nasr et al., 2019, "Comprehensive Privacy Analysis of Deep Learning: Passive and Active White-box Inference Attacks against Centralized and Federated Learning," 2019 IEEE Symposium on Security and Privacy

Antoniou et al., 2018, "Data Augmentation Generative Adversarial Networks", (ICLR 2018 Workshop)

Hariharan & Girshick., 2017, "Low-shot Visual Recognition by Shrinking and Hallucinating Features", (ICCV 2017)

Wang et al., 2018, "Low-shot learning from imaginary data" (CVPR 2018)



_____
### POST-REBUTTAL
_____

After a detailed discussion with the authors, I decided to increase my original rating from 3 to 7. The initial low rating was due to initially hidden assumptions and a poorly defined scope of data privacy which are central in the paper. These have been discussed and clarified by the authors during the rebuttal. The authors also addressed my concerns regarding the novelty and source-domain leakage into the intermediate domain. The authors have agreed to improve the clarity, literature review, dampen down on the privacy claims, and include additional experiments, and I am happy to increase the rating. Although the privacy claims are now not as strong as originally claimed (e.g. the method does not guard against source-information leakage, but rather shelters individual source data points from a possible data leakage), the paper still opens up an interesting area of research and presents a novel method that will likely attract attention in the community.



**Time Spent Reviewing:**

15

---

> ### Author Response · Authors · 2021-08-09
> **Response to other comments**
>
> >Q3. Lack of sufficient evidence for no source domain features in intermediate data.
>
> A3: We have calculated the empirical maximum mean discrepancy (MMD [r1]) value between source data and generated intermediate data, and the results show that the discrepancy between source data and intermediate data is very large compared to the MMD value between any two subsets of the source data. Therefore, we argue that intermediate data are very different from the source data. Moreover, we use the recent MMD test [r2] to test if source data and intermediate data are from the same distribution, and the results show that source data and intermediate data are significantly different in the view of statistics. In the revision, we will add the above MMD results into the main text and illustrate the generated images and source images in the Appendix (we can easily see that they are very different).
>
> >Q4. The relation between FHA and FSL is poorly explained, and FSL is poorly cited.
>
> A4: The main difference between FHA and FSL is the knowledge of source domain. For FHA, the knowledge of source domain is a well-trained classifier. While, for FSL, the knowledge of source domain is many labeled source data or weaker form (e.g., pairwise similarity). In addition, we will cite more FSL papers according to your suggestions.
>
> >Q5. The paper is poorly written.
>
> A5: Thanks for your sincere reminder. We will check the grammar issues and spelling errors carefully and modify them in the revised version. There are some acronyms and mathematical notation in our paper, and they may make readers confused. We will add a table to summarize these acronyms and mathematical notation in the revised version.
>
> >Q6. The TOHAN improvements over baselines are mostly marginal.
>
> A6: The FHA we proposed is a novel and hard problem. As can be seen from Table 1, TOHAN outperforms other baselines significantly. As some baselines (e.g., S+F and T+F) are  ablated version of TOHAN, the improvement is not very large but already remarkable for such a hard problem. As also can be seen from Figure 3, compared with other existing method (i.e., DAPN), TOHAN realizes a great improvement.
>
> References
>
> [r1] Gretton, Arthur, et al. "A kernel two-sample test." JMLR, 2012.
>
> [r2] Liu, Feng, et al. "Learning deep kernels for non-parametric two-sample tests." ICML, 2020.

---

> > ### Comment · Reviewer_1Jpd · 2021-08-13
> > **Still not convinced by evidence for no source domain features in intermediate data.**
> >
> > I thank the authors for their willingness to adapt and improve the paper.
> >
> > > We have calculated the empirical maximum mean discrepancy
> >
> > Reporting the MMD calculation might be insufficient. Specifically, a large MMD score is in line with expectation - the average difference between the intermediate and source domains is expected to be large -- after all, the intermediate dataset is not explicitly trained to look like the source data. Moreover, high MMD score does not eliminate the possibility of some overlap between two distributions. What could be more convincing is showing top N (eg. N > 10) most visually similar intermediate-source sample pairs, and showing that they are sufficiently different -- having 4 randomly selected samples from the intermediate distribution as shown in Figure 6 a/b is not enough.

---

> > > ### Author Response · Authors · 2021-08-15
> > > **Response to “Still not convinced by evidence for no source domain features in intermediate data.”**
> > >
> > > > Reporting the MMD calculation might be insufficient.
> > >
> > > Thanks for the constructive comments! Calculating MMD can show that the data we generate, in statistics, are quite different from the source domain. We admit that this distribution-based measure might not be sufficient. Thus, according to your suggestions, we will calculate the PSNR [r1, r2] score to select the top N (e.g., N > 10) most visually similar intermediate samples (compared to source samples) and visually see if intermediate samples and source samples are sufficiently different.
> > >
> > > We want to mention that, there are no source samples in our setting, thus there are no intermediate-source sample pairs (like you said in your comments). We will calculate the PSNR [r1, r2] score between each intermediate sample and all source samples to select the most similar source sample to that intermediate sample. Thus, we can obtain PSNR scores of all intermediate samples, and select the top N samples to visualize them (and the most similar source samples corresponding to them).
> > >
> > > Since we need to rerun the experiment to save the intermediate data, we will report the results when done. We welcome your comments on this experimental setting.
> > >
> > > References
> > >
> > > [r1] A. Horé and D. Ziou, "Image Quality Metrics: PSNR vs. SSIM," ICPR, 2010.
> > >
> > > [r2] Yin, Hongxu, et al. "See through Gradients: Image Batch Recovery via GradInversion." CVPR. 2021.

---

> > > ### Author Response · Authors · 2021-08-19
> > > **Experiment regarding the similarity between intermediate samples and source samples.**
> > >
> > > Thanks for your suggestions. We have completed the suggested experiments.
> > >
> > > Taking MNIST-->SVHN as an example, we have calculated the PSNR values between each intermediate sample and all source samples. We select the intermediate samples that are the most similar to source samples. The **larger** PSNR value indicates the two images are **more similar**. Therefore, we report the top-10 maximal PSNR values below.
> > >
> > > 17.89510727, 17.89479637, 17.89478683, 17.89469147, 17.89465141, 17.89436531,
> > > 17.8941021,  17.89398003, 17.89396667, 17.8939476
> > >
> > > These 10 PSNR values represent the intermediate samples that are the top-10 most similar to source samples. In addition, we also compute the PSNR values of (source, source), (target, target), (source, target), and (intermediate, target) for comparison. We report the average PSNR values of the above four cases.
> > >
> > > (source, source): 52.3372
> > >
> > > (target, target): 54.3005
> > >
> > > (source, target): 9.3209
> > >
> > > (intermediate, target): 50.1722
> > >
> > > As can be seen from the results above, intermediate samples are much closer to the target samples, and they are very different from the source samples. The average PSNR value between source samples and target samples is 9.3209. The top-10 maximal PSNR values between each intermediate sample and all source samples reported above (~17.89) are obviously smaller than 50.1722 (average PSNR value between intermediate samples and target samples). Through this result, we can state that intermediate samples are similar to target samples and very different from source samples. We also visualize the top-10 similar intermediate samples and find that their style looks like SVHN and is very different from MNIST (sorry we cannot put images here).
> > >
> > > Since we cannot put images here, we use the results from Table 2 in [r1] to understand what “PSNR is about 17.89” stands for. The EMI, the worst model-inversion attack baseline, achieves the PSNR value of more than 18.69 (more than 20.00 in most cases). Then, from Figure 2, we can find the images generated by EMI are still noisy. Their results mean that  “PSNR is more than 18.69” represents that the generated images do not contain useful private information.
> > >
> > > Therefore, the above evidence shows that the generated intermediate data contain no source-domain features, and the privacy in the source domain does not leak when generating the intermediate data.
> > >
> > > We wonder if you are satisfied with our experimental analysis above, and you are welcome to propose more valuable comments. We also wish for your suggestions to improve the quality of our paper further.
> > >
> > > References
> > >
> > > [r1] Zhang, Yuheng, et al. "The secret revealer: Generative model-inversion attacks against deep neural networks." CVPR, 2020.

---

> > > > ### Comment · Reviewer_1Jpd · 2021-08-24
> > > > **This is helpful.**
> > > >
> > > > The experiments are very helpful. Thank you.

---

> ### Author Response · Authors · 2021-08-09
> **Response to “the novelty seems limited”**
>
> >Q2 (the novelty seems limited). As far as I am aware, the paper also proposes a novel method - TOHAN - which is a minor adaptation of FADA [23] into the setting. The method architecture seems to be heavily inspired by FADA [23]. Unlike FADA, TOHAN generates and uses the generated intermediate domain instead of the original source domain. However, apart from that, it is not entirely clear what the technical differences are between these methods. Which line in Algorithm 1 would be different for FADA / S-FADA / T-FADA / ST-FADA?
>
> A2. As other reviewers point out, the novelty of our paper contains two parts: 1) the proposed setting is novel and interesting and 2) the proposed solution (generating data to solve the FHA) is very new to the field. The previous HTL methods (the most related methods) only consider proposing a new loss function [r1] or finding/digging relation among available data and few-shot data (Reviewers 4XQu and 6L9x). However, our solution looks novel, reasonable and interesting (Reviewers FhwF, 4XQu and 6L9x). Thus, it can be seen that our main technical contribution is to propose to generate and use the generated intermediate domain instead of the original source domain, which is a novel perspective to the field. Since our major contribution is not to propose a good measure to show the discrepancy between two domains, we simply adopt the measure used in FADA [23].
>
> Compared to FADA, the difference lies in line 1-5. The setting is different. For FADA, they can access many labeled source data, and they use them to train a model. For TOHAN, we only can access a source model, and we use it and target data to generate intermediate data.
>
> Compared to S-FADA, the difference lies in line 5. The loss function of generators is different. For S-FADA, the loss function is Eq. (5). While, for TOHAN, the loss function is Eq. (7).
>
> Compared to T-FADA, the difference lies in line 5. The loss function of generators is different. For T-FADA, the loss function is Eq. (6). While, for TOHAN, the loss function is Eq. (7).
>
> Compared to ST-FADA, the difference lies in line 2-12. ST+FADA is a two-step approach.  In detail, it first generates intermediate data (line 2-5), then pretrains group discriminator (line 6-8), and updates target model (encoder & classifier) and group discriminator alternatively (line 9-12) as last. However, TOHAN is the one-step version of ST-FADA. In detail, it first pretrains the generators for 100 epochs (line 2-5) and pretrains the group discriminator for 50 epochs, and then it updates generators (line 2-5), target model and group discriminators  (line 9-12) simultaneously at each epoch.
>
> Reference
>
> [r1] Kuzborskij, Ilja. Theory and algorithms for hypothesis transfer learning. No. THESIS. EPFL, 2018.

---

> ### Author Response · Authors · 2021-08-09
> **A factual error contained in the comments**
>
> >Q1. There might be a possible fundamental flaw to the claim of full data privacy.
>
> A1 (factual error contained in the comments): We have read the literature you provided carefully, and, unfortunately, find that there is a **factual error** in your comments.
>
> We argue that it is meaningless to talk about the privacy leakage when you are ignoring the capability of the adversary attackers (who want to illegally obtain the data).
>
> In the suggested paper, Nasr et al. (2019), they assume that the adversary attackers can access a well-trained classifier (like we have in FHA) and a dataset that contains the training data (we do not have them in FHA). The aim of attackers is to identify the training data accurately from the dataset. In our paper (in FHA), source-data owners only provide a well-trained classifier, which is a totally different scenario from that considered by Nasr et al. (2019). Generally speaking, attackers assumed in Nasr et al. (2019) have capability to access the classifier and the data, while the potential attackers in FHA only have the capability to access the classifier.
>
> In the literature, Ateniese et al. (2015), they did not focus on the data leakage since they cared about the public data (see the first paragraph on the third page of Ateniese et al. (2015)). Specifically, they mentioned that "we are not interested in privacy leaks, but rather in discovering anything that makes classifiers better than others". In fact, the information that they can glean from comparing multiple classifiers is that, e.g., if model A performs better than model B but they're known to be implementing the same algorithm, then model A's training data must be more valuable (in some sense) than that of model B.  (an example they provide is that: the majority of training samples came from female voices or from voices of people with marked accents). It is clear that they are talking about different things from ours. In their setting, they do not try to attack the data privacy.
>
> Then, based on the research line of privacy attacks, we found the most relevant attack is the model-inversion attack, which aims to recover the input images using a well-trained classifier [r1]. However, even in the recent work [r1], they still need to access auxiliary data to help recover input images. For example, in [r1], they actually need to access images whose background is similar to the training data (please see the configurations of experiments of [r1]), which is very different from our problem setting. In FHA, we can only access very few target data whose background is very different from the source data (e.g., MNIST as the source domain, and SVHN as the target domain).
>
> Thus, no matter which attacks in the literature are considered (attacks you mentioned in Nasr et al. (2019) or model-inversion attacks we found), the source-data privacy is protected well in our setting. The main reason is that, in our setting FHA, attackers can only obtain a well-trained classifier which is a very restricted condition for the existing attacks in the literature. In the literature, hypothesis transfer learning methods also claimed that they can protect the source-data privacy because they only need to access the source model rather than the source data [r2].
>
> Moreover, even in the future, there will be attacks that can recover input images from a well-trained classifier, the data owners will know these attacks and will only provide us a classifier that can defend against such attacks, which is not the scope of our paper.
>
> Since your comments contain this factual error (which, of course, influences your judgments regarding our paper), we sincerely hope you can reconsider the major contributions of our paper. We will also include the above discussion in the main text of our paper, which is a good supplement to our setting and our solution.
>
> References
>
> [r1] Zhang, Yuheng, et al. "The secret revealer: Generative model-inversion attacks against deep neural networks." CVPR, 2020.
>
> [r2] Liang, Jian, et al. "Do we really need to access the source data? source hypothesis transfer for unsupervised domain adaptation." ICML, 2020.

---

> > ### Comment · Reviewer_1Jpd · 2021-08-13
> > **Still doubtful about the data privacy claims**
> >
> > I thank the authors for the clarifications. While I agree with some points made, I still have doubts about the data privacy claims in the main paper.
> >
> > > Nasr et al. (2019) have capability to access the classifier and the data
> >
> > I agree that Nasr et al. (2019) assume that an adversary has access to at least some source-domain data, so it is less applicable in this setting (except maybe when an adversary gains hold of some source data through data leaks/hacking - which is a general risk and not necessarily to do with the proposed TOHAN model.)
> >
> > > Ateniese et al. (2015) ... did not focus on the data leakage
> >
> > I partially agree about Ateniese et al. (2015). Specifically, they aim to extract information that is not necessarily about individual data points but rather general source-domain information that could help improve the performance of classifiers and give a competitive advantage to vendors. **However, such information could still be considered private/sensitive/useful**. I do not see why this type of attack cannot be mounted against TOHAN to extract "useful" information.
> >
> > > we found the most relevant attack is the model-inversion attack
> >
> > The work by Zhang et al. (2020) - i.e. [r1] as suggested by the authors - is relevant and appropriate to the discussion and should be referred to in the main paper. However, I am not convinced why the Paper2423 authors think this type of attack does not apply to their setting. The assumption that an adversary has access to some auxiliary data similar to the source-domain is not unrealistic. In fact, I would argue that an adversary is likely to have similar datasets for some domains (e.g. face recognition datasets are easily available online, as argued by Zhang et al. (2020) in section 1, paragraph 4). Therefore, this type of attack could still be used against TOHAN and leak private/sensitive information in these situations. The assumption that an adversary does not have access to similar data can be easily broken in the real world and undermines the privacy claim in the main paper.
> >
> > > For example, in [r1], they actually need to access images whose background is similar to the training data
> >
> > I assume that "background" refers to "Possible Auxiliary Knowledge" in section 3.1 of Zhang et al. (2020). However, in section 5.3.1, several experiments are carried out, including one where the auxiliary knowledge is **not** used by the attacker (see Table 1). GMI achieves well above random according to the Attack Accuracy metric (in the 1000-class problem), suggesting that a significant amount of information can still leak with the attacker having access to the classifier alone and easily available data on the internet.
> >
> > *I remain unconvinced by the authors' response.* The central claim that the TOHAN will not leak any "privacy information" and "useful knowledge" (line 245-247) seems contrary to work by Zhang et al. (2020) and Ateniese et al. (2015) that show that it is possible to extract such information with access to the classifier alone and/or easily accessible data available online.

---

> > > ### Author Response · Authors · 2021-08-16
> > > **Response to “Still doubtful about the data privacy claims” -- part 2.**
> > >
> > > In the rest of responses, we reply to your new comments in detail.
> > >
> > > >I agree that Nasr et al. (2019) assume that an adversary has access to at least some source-domain data, so it is less applicable in this setting (except maybe when an adversary gains hold of some source data through data leaks/hacking - which is a general risk and not necessarily to do with the proposed TOHAN model.)
> > >
> > > Thanks for replying to our responses. Nasr et al. (2019) indeed need to access other data that are not available in our setting.
> > >
> > >
> > > >I partially agree about Ateniese et al. (2015). Specifically, they aim to extract information that is not necessarily about individual data points but rather general source-domain information that could help improve the performance of classifiers and give a competitive advantage to vendors. However, such information could still be considered private/sensitive/useful. I do not see why this type of attack cannot be mounted against TOHAN to extract "useful" information.
> > >
> > > Each model contains some information, and it is not appropriate to talk about the privacy leakage without thresholds. The privacy considered in our paper and [r1] is limited to data input leakage that is not relevant to Ateniese et al. (2015).
> > >
> > > What we will do to improve our paper:
> > >
> > > We will clearly restate what privacy we can protect (we can protect the leakage of source data) and discuss the privacy protection more in the revision, like existing literature regarding the privacy attack.
> > >
> > >
> > > >In fact, I would argue that an adversary is likely to have similar datasets for some domains (e.g. face recognition datasets are easily available online, as argued by Zhang et al. (2020) in section 1, paragraph 4). Therefore, this type of attack could still be used against TOHAN and leak private/sensitive information in these situations. The assumption that an adversary does not have access to similar data can be easily broken in the real world and undermines the privacy claim in the main paper.
> > >
> > > We argue that the owner of source data will not let their data be similar to images on the internet (for the private concern). When talking about privacy leakage, we need to notice that the data owner has a strong will to protect the data. Thus, it is very difficult for an adversary to access similar images to the original data (unless the adversary hacked into the database of the owner, which is a general risk and not necessarily to do with the proposed TOHAN model).
> > >
> > > Although Zhang et al. (2020) use no auxiliary knowledge in Table 1, they still use the public dataset (see “Public knowledge distillation” and Eq. (1) in Section 3.2 in Zhang et al. (2020)). Therefore, they also make use of similar data to the private data. In their experiments, the public data come from the dataset that also contains the private data (rather than the internet data), which can be found in the “Protocol” in Section 5.1 in [r3]. This is a strong assumption hidden in their method and experiments. In our setting, however, accessing such public data is unrealistic, since source-data owners will not let their data be similar to public data.
> > >
> > >
> > > >However, in section 5.3.1, several experiments are carried out, including one where the auxiliary knowledge is not used by the attacker (see Table 1).
> > >
> > > Although they use no auxiliary knowledge in Table 1, they still use the public dataset (see “Public knowledge distillation” and Eq. (1) in Section 3.2 in Zhang et al. (2020)). Therefore, they also make use of similar data to the private data. In their experiments, the public data come from the dataset that also contains the private data (rather than internet data), which can be found in the “Protocol” in Section 5.1 in their paper. This is a strong assumption hidden in their method and experiments. In our setting, however, accessing such public data is unrealistic, since source-data owners will let their data be similar to public data.
> > >
> > > Besides, there are only toy examples used in [r3]. For a real privacy-protection scenario, such as the pathologic image data, hospitals will not make such data public (for privacy protection). In addition, to be safer, the source model can be adversarially trained against the model-inversion attacks [r2], and such model-inversion attacks will fail further. However, this is beyond the scope of our paper (like we discussed in “The general response” in this thread).
> > >
> > > >A significant amount of information can still leak with the attacker having access to the classifier alone and easily available data on the internet.
> > >
> > > We argue that the owner of source data will not let their models be trained with images that are similar to images on the internet. When talking about privacy leakage, we need to notice that the data owner has a strong will to protect the data. Thus, it is very difficult for an adversary to access similar images to the original data (unless the adversary hacked into the database of the owner, which is a general risk and not necessarily to do with the proposed TOHAN model). In [r3], their public data and private data are actually from the same dataset, which is a strong implicit assumption in their method and experiments.
> > >
> > > References
> > >
> > > [r1] Liang, Jian, et al. "Do we really need to access the source data? source hypothesis transfer for unsupervised domain adaptation." ICML, 2020.
> > >
> > > [r2] Titcombe, Tom, et al. "Practical Defences Against Model Inversion Attacks for Split Neural Networks." ICLR Workshop on DPML, 2021.
> > >
> > > [r3] Zhang, Yuheng, et al. "The secret revealer: Generative model-inversion attacks against deep neural networks." CVPR, 2020.

---

> > > > ### Comment · Reviewer_1Jpd · 2021-08-24
> > > > **Very helpful comments.**
> > > >
> > > > The latest comments from the authors help to clarify the scope of data protection. I am willing to increase my score to 7, provided that the authors can include the discussion on data privacy and list the hidden assumptions that guarantee the claimed data protection. Specifically, the following, in my opinion, should be included in the final version:
> > > >
> > > > - Clearly state what information can be protected by the model (e.g. leakage of individual source data points).
> > > > - Clearly state what information cannot be protected by the model (e.g. partial or general information about the source domain that could be extracted using methods such as Ateniese et al. (2015))
> > > > - Clearly state the assumptions made that, if broken, could lead to loss of source-domain protection (e.g. that the source-domain data is sufficiently different to any data that is available to an attacker - such as data from the Internet - otherwise, Zhang et al. (2020) could be used against the proposed method).
> > > > - Include the experiments from the other comment - showing that intermediate data is sufficiently different from the source domain
> > > > - Include an appropriate literature review on data privacy and model inversion attacks in the related works section.
> > > > - Fix the literature review on Few-Shot Learning (as discussed in my original review)
> > > > - Grammar issues and spelling errors are resolved
> > > >
> > > > With these corrections and the author's helpful clarifications about the model's novelty made in the other comment, I believe this is can be an interesting and relevant paper for the community.

---

> > > > > ### Author Response · Authors · 2021-08-24
> > > > > **Many thanks for your constructive suggestions!**
> > > > >
> > > > > >The latest comments from the authors help to clarify the scope of data protection. I am willing to increase my score to 7, provided that the authors can include the discussion on data privacy and list the hidden assumptions that guarantee the claimed data protection. Specifically, the following, in my opinion, should be included in the final version.
> > > > >
> > > > > **General Response:** Many thanks for your constructive comments that indeed improve the clarity and quality of our paper! **We will follow all of your suggestions to revise our paper.** In the following, please allow us to demonstrate how our discussions will be included in the final version to follow your suggestions.
> > > > >
> > > > >
> > > > > > Clearly state what information can be protected by the model (e.g. leakage of individual source data points).
> > > > > > Clearly state what information cannot be protected by the model (e.g. partial or general information about the source domain that could be extracted using methods such as Ateniese et al. (2015))
> > > > >
> > > > > **Answer 1 (A1):** After the “privacy” appears in the introduction, we will add a follow-up paragraph to demonstrate that we mainly can protect the source-data leakage rather than the information leakage. In this paragraph, we will discuss the aim of Ateniese et al. (2015) and show studies related to information leakage. Such a  paragraph will avoid misleading readers to think about the information leakage (like Ateniese et al. (2015) did). Meanwhile, the paragraph “Why does TOHAN prevent the leakage of private information effectively?” (lines 240-247) will be removed (to avoid misleading readers).
> > > > >
> > > > > > Clearly state the assumptions made that, if broken, could lead to loss of source-domain protection (e.g. that the source-domain data is sufficiently different to any data that is available to an attacker - such as data from the Internet - otherwise, Zhang et al. (2020) could be used against the proposed method).
> > > > >
> > > > > **Answer 2 (A2):** In the problem setting, we will show that, in FHA, the source-data owners need to ensure that source-domain data is sufficiently different from the data that is available to an attacker (i.e., the assumption) - such as data from the Internet - otherwise, the model-inversion attack methods (e.g., Zhang et al. (2020)) could be used to leak source data.
> > > > >
> > > > > Then, we will add a paragraph (after the problem setting) to review the literature regarding the model-inversion attacks (including Zhang et al. (2020)). In this paragraph, we will discuss when the source data can be recovered (namely, the source data may leak). We will also suggest that the source-data owners might utilize the defending techniques (against the model-inversion attacks) to train their source model.
> > > > >
> > > > > >Include the experiments from the other comment - showing that intermediate data is sufficiently different from the source domain
> > > > >
> > > > > **Answer 3 (A3):** We will add the experiments you suggest into our experimental section as an important subsection in our experiment section (we have completed such experiments and reported results in our previous response. In this subsection, we will also include the figures we generated, which can greatly help readers see the difference between the intermediate data and the source data). This subsection will show that intermediate data is sufficiently different from the source domain and the source data is not leaked during TOHAN.
> > > > >
> > > > > >Include an appropriate literature review on data privacy and model inversion attacks in the related works section.
> > > > >
> > > > > **Answer 4 (A4):** We will add a paragraph (after the problem setting) to review the literature regarding the model-inversion attacks (including Zhang et al. (2020)). In this paragraph, we will discuss when the source data can be recovered (namely, the source data may leak). After this paragraph, we will review the literature regarding the data privacy attacks to thoroughly discuss what information we may leak and what information/data we can protect (as an extension of the newly-added paragraph in **A1**).
> > > > >
> > > > > >Fix the literature review on Few-Shot Learning (as discussed in my original review)
> > > > >
> > > > > **Answer 5 (A5):** According to your suggestions, we will compare the FHA and FSL in detail. We will also cite FSL papers carefully according to your comments. Specifically, the papers you suggested ((Antreas et al., 2018), (Hariharan & Girshick., 2017), (Wang et al., 2018)) will be reviewed, and we will demonstrate the main difference between the suggested papers and our paper.
> > > > >
> > > > > >Grammar issues and spelling errors are resolved
> > > > >
> > > > > **Answer 6 (A6):** We will fix all grammar issues you find and will seek a native English speaker to thoroughly revise the whole paper.
> > > > >
> > > > > >With these corrections and the author's helpful clarifications about the model's novelty made in the other comment, I believe this can be an interesting and relevant paper for the community.
> > > > >
> > > > > **General Response:** We are glad to discuss our paper with you, which is a very constructive and valuable improvement for our paper. If you have more suggestions, please kindly let us know. We are very glad to take them further.

---

> > > ### Author Response · Authors · 2021-08-16
> > > **Response to “Still doubtful about the data privacy claims” -- part 1.**
> > >
> > > >(The general concern) The central claim that the TOHAN will not leak any "privacy information" and "useful knowledge" (line 245-247) seems contrary to work by Zhang et al. (2020) and Ateniese et al. (2015) that show that it is possible to extract such information with access to the classifier alone and/or easily accessible data available online.
> > >
> > > **The general response**: Thanks for your responses. We want to point out that Zhang et al. (2020) need to use public data to train a good image generator (see “Public knowledge distillation” and Eq. (1) in Section 3.2 in Zhang et al. (2020)). However, we do not have such public data in our setting. Actually, the public data used in Zhang et al. (2020) are from the dataset that also **contains the private data** (they split a well-known dataset into public part and private part, see “Protocol” in Section 5.1 in their paper!), meaning that their experiments are built on a **strong assumption**: public data and private data come from the same dataset and thus they share the same style, i.e., they are from the same meta-distribution [r1, r2]. It is unrealistic since source-data owners will not let their data be similar to public data if they care about the privacy issues.
> > >
> > > There might also be a misunderstanding about the “privacy leakage of TOHAN”. The privacy protection of TOHAN is the passive protection (same as the SHOT [r3]) rather than the active protection, because that we, as a user of the source model, do not have a will to attack the model to steal information from it (i.e., there are no well-motivated adversaries in our setting). In our setting, the source model is only available to us, which means that the adversary must be one of us. However, our aim is to train a good target-domain classifier rather than attack the source model, and attacking the source model might be less beneficial than training a good target-domain classifier in the real world (i.e., the motivation to attack might not exist).
> > >
> > > Besides, in the training process of TOHAN, we are not trying to generate source data since our aim is to train a classifier for the target domain (we are conducting the experiments you suggested and will report back when done).
> > >
> > > Even in the future, there will be attacks that can successfully recover input data from a well-trained classifier (note that existing methods cannot do this), the data owners will know these attacks and will only provide us a classifier that can defend against such attacks, which is not the scope of our paper.
> > >
> > > If we want to protect the source information actively (i.e., the active protection), there will be another condition in our problem setting: to be aware that one of us is a potential adversary (this might be an extreme scenario in the real world because attacking the source model might be less beneficial than training a good target-domain classifier). To solve this new problem, we need to design a new method to defend against the model-inversion attacks (by referring to, e.g., [r4]). However, it is not the scope of our paper either.
> > >
> > > Since 1) there are no existing method that can successfully recover images only using a classifier; and 2) there are no well-motivated adversaries in our setting; and 3) the training process of TOHAN will not generate the source data (we are verifying this point according to your suggestions); and 4) we can only access the source model, we do not leak the source data.
> > >
> > > Except for the passive privacy-protection benefit of TOHAN, **please allow us to explain other contributions of our paper**:
> > >
> > > 1) We can train a good target-domain classifier only with a classifier and few target data, which is more efficient than the previous FDA methods where abundant source data will be involved in the training (lines 49-51).
> > >
> > > 2) The proposed solution (generating data to solve the FHA) is very new to the field. The previous HTL methods (the most related methods) only consider proposing a new loss function [r3] or finding/digging relation among available data and few-shot data (Reviewers 4XQu and 6L9x). However, our solution looks novel, reasonable and interesting (Reviewers FhwF, 4XQu and 6L9x). Our main technical contribution is to propose to generate and use the generated intermediate domain instead of the original source domain, which is a novel perspective to the field.
> > >
> > > We argue that the privacy-protection property (passive way to protect) of TOHAN is only one of our contributions. Our technical contribution is also novel to the field (as pointed out by other reviewers).
> > >
> > > What we will do to improve our paper:
> > >
> > > In the revision, we will explicitly state what kind of privacy leakage (actually, the data leakage) we can protect and discuss the literature you provided and references we found regarding the model-inversion attacks. We will also illustrate the difference between the generated data and the source data according to your constructive comments. After the detailed discussion regarding the privacy-protection ability of TOHAN, future readers will not misunderstand our setting and solution. What do you think about these changes? If we add these discussions, are your major concerns addressed well? We sincerely hope for your future comments regarding improving the quality of our paper.
> > >
> > > References
> > >
> > > [r1] Snell, Jake, Kevin Swersky, and Richard S. Zemel. "Prototypical networks for few-shot learning." NeurIPS, 2017.
> > >
> > > [r2] Finn, Chelsea, Pieter Abbeel, and Sergey Levine. "Model-agnostic meta-learning for fast adaptation of deep networks." ICML, 2017.
> > >
> > > [r3] Liang, Jian, et al. "Do we really need to access the source data? source hypothesis transfer for unsupervised domain adaptation." ICML, 2020.
> > >
> > > [r4] Titcombe, Tom, et al. "Practical Defences Against Model Inversion Attacks for Split Neural Networks." ICLR Workshop on DPML, 2021.

---

> ### Author Response · Authors · 2021-08-26
> **Thank Reviewer 1Jpd for considering improving the score from 3 to 7!**
>
> Dear Reviewer 1Jpd,
>
> Thanks for considering improving your score from 3 to 7 in your last comment (https://openreview.net/forum?id=vrkQ07gp0kq&noteId=kso3LW6hVUa). **We will follow all of your suggestions.** Since we find that our reply is folded by the system, please allow us to paste our reply in this new thread (see the following responses).
>
> >The latest comments from the authors help to clarify the scope of data protection. I am willing to increase my score to 7, provided that the authors can include the discussion on data privacy and list the hidden assumptions that guarantee the claimed data protection. Specifically, the following, in my opinion, should be included in the final version.
>
> **General Response:** Many thanks for your constructive comments that indeed improve the clarity and quality of our paper! We will follow all of your suggestions to revise our paper. In the following, please allow us to demonstrate how our discussions will be included in the final version to follow your suggestions.
>
> > Clearly state what information can be protected by the model (e.g. leakage of individual source data points).
> > Clearly state what information cannot be protected by the model (e.g. partial or general information about the source domain that could be extracted using methods such as Ateniese et al. (2015))
>
> **Answer 1 (A1):** After the “privacy” appears in the introduction, we will add a follow-up paragraph to demonstrate that we can mainly protect the source-data leakage rather than the information leakage. In this paragraph, we will discuss the aim of Ateniese et al. (2015) and show studies related to information leakage. Such a  paragraph will avoid misleading readers to think about the information leakage (like Ateniese et al. (2015) did). Meanwhile, the paragraph “Why does TOHAN prevent the leakage of private information effectively?” (lines 240-247) will be removed (to avoid misleading readers).
>
> > Clearly state the assumptions made that, if broken, could lead to loss of source-domain protection (e.g. that the source-domain data is sufficiently different to any data that is available to an attacker - such as data from the Internet - otherwise, Zhang et al. (2020) could be used against the proposed method).
>
> **Answer 2 (A2):** In the problem setting, we will show that, in FHA, the source-data owners need to ensure that source-domain data is sufficiently different from the data that is available to an attacker (i.e., the assumption) - such as data from the Internet - otherwise, the model-inversion attack methods (e.g., Zhang et al. (2020)) could be used to leak source data.
>
> Then, we will add a paragraph (after the problem setting) to review the literature regarding the model-inversion attacks (including Zhang et al. (2020)). In this paragraph, we will discuss when the source data can be recovered (namely, the source data may leak). We will also suggest that the source-data owners might utilize the defending techniques (against the model-inversion attacks) to train their source model.
>
> >Include the experiments from the other comment - showing that intermediate data is sufficiently different from the source domain
>
> **Answer 3 (A3):** We will add the experiments you suggest into our experimental section as an important subsection in our experiment section (we have completed such experiments and reported results in our previous response. In this subsection, we will also include the figures we generated, which can greatly help readers see the difference between the intermediate data and the source data). This subsection will show that intermediate data is sufficiently different from the source domain and the source data is not leaked during TOHAN.
>
> >Include an appropriate literature review on data privacy and model inversion attacks in the related works section.
>
> **Answer 4 (A4):** We will add a paragraph (after the problem setting) to review the literature regarding the model-inversion attacks (including Zhang et al. (2020)). In this paragraph, we will discuss when the source data can be recovered (namely, the source data may leak). After this paragraph, we will review the literature regarding the data privacy attacks to thoroughly discuss what information we may leak and what information/data we can protect (as an extension of the newly-added paragraph in **A1**).
>
> >Fix the literature review on Few-Shot Learning (as discussed in my original review)
>
> **Answer 5 (A5):** According to your suggestions, we will compare the FHA and FSL in detail. We will also cite FSL papers carefully according to your comments. Specifically, the papers you suggested ((Antreas et al., 2018), (Hariharan & Girshick., 2017), (Wang et al., 2018)) will be reviewed, and we will demonstrate the main difference between the suggested papers and our paper.
>
> >Grammar issues and spelling errors are resolved
>
> **Answer 6 (A6):** We will fix all grammar issues you find and will seek a native English speaker to thoroughly revise the whole paper.
>
> >With these corrections and the author's helpful clarifications about the model's novelty made in the other comment, I believe this can be an interesting and relevant paper for the community.
>
> **General Response:** We are glad to discuss our paper with you, which is a very constructive and valuable improvement for our paper. If you have more suggestions, please kindly let us know. We are very glad to take them further.
>
> Best regards,
>
> Authors of Paper2423

---

### Official Review · Reviewer_6L9x · 2021-07-14

**Rating:** 7
**Confidence:** 4

**Summary:**

This paper focuses on a problem setting where data are not available but classifiers are. The possible benefits of this setting lie in the privacy reservation, i.e., data used in previous few-shot learning may leak the private information. The proposed solution does not use a fine-tuning-based method but a sample-generation-based method, which is motivated by the learnability of semi-supervised learning.

**Limitations And Societal Impact:**

The limitations and potential negative societal impact of their work have been discussed well.

**Main Review:**

The proposed solution thought few-shot learning in a different way. The most of current methods address few-shot learning via finding/digging relation among available data and few-shot data. Data can be from different distributions (although they should be related) [r1]. In this paper, generating samples for the target domain is proposed and validated. If the samples are suitable for the target domain, a good classifier should be achievable (like SSL).

Major concerns/questions:

   1. Although generating samples look novel and interesting, how to ensure a reliable generating process is unclear to me. There are many ways to influence the process and make the generated sample unreliable. Regarding this concern, I am not sure why the source domain can help as well. If the source cannot help, the generated samples must be unreliable, is that right?

   2. I notice that HTL might be a very similar setting to the proposed setting. What are the major differences between both? Have you tried to compare TOHAN with HTL methods?

   3. A large pre-trained classifier also seems suitable for solving the proposed setting. Such a classifier contains enough knowledge that is useful for the target domain as well. If so, what is the meaning to consider the proposed setting and solution?

  4. Since you use a generator (DC-GAN) for each class, I was worried about the computational cost when scaling the method to larger datasets. 10 generators might be acceptable, but when the number becomes 200 or even 1000, is there any way to alleviate the computational cost?

  5. I do not understand $\mathcal{H}_{P,X}$ in Eq. (2). How is it calculated? Also, what is the meaning of the $t$ in the definition of Theorem 1? I guess it is some kind of form/assumption but I think the authors should elaborate on it here.  The subscript of $M$ in line 71 appears abruptly without any description.

  6. Why do you introduce M in Eq. (6)? Does this M influence the final results?

[r1] Model-agnostic meta-learning for fast adaptation of deep networks. ICML 2017.

----
## Post-Rebuttal Update
----

After viewing the rebuttal and other reviewers' comments, I feel more confident in the authors' methods. I think the authors did a great job in addressing my questions, so I have decided to raise my score to 7. There is something the authors need to address more but I do think it does not diminish the contribution of this work.

**Time Spent Reviewing:**

3

---

> ### Author Response · Authors · 2021-08-09
> **Response to Reviewer 6L9x**
>
> Thanks for your constructive comments! We will answer your major concerns in the following.
>
> >Q1. If the source cannot help, the generated samples must be unreliable, is that right?
>
> A1: Yes, we totally agree with you. This problem has been studied in Ablation Study. We can find that generating samples only with target data (‘T+F’ in Table 3) performs worse than generating samples with source model and target data (‘TOHAN’ in Table 3). The reason is simple, as we cannot learn the whole target distribution with very limited target data. Therefore, the data generated by this way lacks diversity.
>
> >Q2. What are the major differences between FHA and HTL? Have you tried to compare TOHAN with HTL methods?
>
> A2: We compare HTL and FHA in line 105-118 in our paper. The difference between FHA and HTL is that FHA requires less labeled target data (e.g., one sample per class).
> We compare TOHAN and dkdHTL [41] (a novel HTL method) in Appendix F. In the revision, we will discuss the relation between FHA and HTL in detail and move the experiment regarding TOHAN and dkdHTL to the main text.
>
> >Q3. A large pre-trained classifier also seems suitable for solving the proposed setting. A large pre-trained classifier also seems suitable for solving the proposed setting and solution?
>
> A3: It is a very good question. Yes, indeed, the large pre-trained classifier seems a good solution to FHA, and we have tried it when we first want to solve the FHA. However, we find that it is very difficult to fine-tune a big model with such a small target domain, and the overfitting situation significantly reduces the accuracy in the target domain. It is the reason why we consider a fine-tuning method (FT in our paper) that is only based on the source model rather than a pre-trained model. From the experiments, it is clear that we have much better performance than the FT method. Based on your comments, we think it is necessary to conclude our previous results regarding the pre-trained models in the main text of the revision.
>
> Besides, training a large scale classifier requires very abundant data. FHA mainly focuses on private scenario (e.g., data from personal phone or surveillance cameras), acquiring very abundant private data is clearly not very realistic.
>
> >Q4. How to alleviate computational cost when the dataset scale increases larger?
>
> A4: When the number of categories increases (e.g., 200 or even more than 1000), we can employ only one generator. Specifically, all the 200 categories share the backbone, and we concatenate 200 heads after the backbone. Each head is used for the generation of one category. Through this way, the number of parameters drops greatly, and the computational cost will not be too much when the dataset scale increases larger.
>
> >Q5. How to calculate $\mathcal{H}_{P,\chi}$ in Eq.(2)? What is the meaning of $t$ in Theorem 1? What is the meaning of superscript $M$ in line 171?
>
> A5: Let $\mathcal{H}_{P,\chi}=\left\\{f\in\mathcal{H}: \chi(f,P)\ge 1-\tau\right\\}$.
>
> We denote $\mathcal{H}_{P,\chi}(\tau)[m,P]$
>
> as the expected value of splits of $m$ data drawn from distribution $P$ using hypotheses in $\mathcal{H}_{P,\chi}$.
>
> In Theorem 1, $t$ is a constant, and we assume $\chi(c^*,P)=1-t$. The larger $t$ represents smaller compatibility between $c^*$ and $P$.
> In line 171, the superscript $M$ is a mark, indicating $G_n^M$ is a matrix.
>
> >Q6. Why do we introduce $M$ in Eq.(6)? Does $M$ influences the final results?
>
> A5: $M$ means the maximal distance between any two data points in feature space $\mathcal{X}$. We introduce $M$ to normalize Eq. (6), then we can restrict the value of Eq.(6) within [0,1], which can be regarded as the compatibility.

---

### Official Review · Reviewer_4XQu · 2021-07-14

**Rating:** 7
**Confidence:** 4

**Summary:**

As the increasing need for privacy protection, previous few-shot DA (FDA) has limited applications in the real world. To overcome this drawback of the FDA, this paper considers a very challenging problem setting, few-shot hypothesis adaptation (FHA), which is the most difficult part of hypothesis transfer learning (HTL). Compared to previous solutions to HTL, the authors argue that generating target samples is a good solution. The proposed method is validated on common datasets.

**Limitations And Societal Impact:**

Yes, the authors have adequately addressed the limitations and potential negative societal impact of their work.

**Main Review:**

Pros:

1.  The considered problem setting is practical and is not solved well in the literature. In the current era, how to protect privacy is quite important. If we can use well-trained classifiers (from other domains) to help obtain a good classifier for the target domain, many more available well-trained classifiers can be used, which can save a lot of energy to train a deep network for the target domain or to train a very big generalized deep network.

2. Compared to existing fine-tuning methods, the proposed solution considers the domain gap between the backbone network and the target domain, which is a research gap in the current literature. Besides, generating unlabeled data for the target domain sounds interesting, especially in few-shot situations.

3. Experiments are conducted on five data sets. Baselines, including FDA methods and HTL methods, are used to validate if the proposed method can handle FHA well.

Cons:

1. In Problem 1, we have two domains: source and target. However, if the source domain is a random domain, can we still obtain a useful classifier for the target domain? The key assumption that makes FHA solvable is missing. What is the key to ensuring that FHA can be solved? This is the major concern about the considered setting.

2. The proposed solution, TOHAN, is a hybrid one that repeats generating and adapting. However, how could you know the initial steps being good? At the initial stage, we might not get good target samples, right? How could we ensure we can get good results from possible bad target samples in the loop mentioned above?

3. SHOT is actually not a classical HTL method. Network-based HTL methods should be also considered as a baseline. Besides, more HTL methods should be reviewed, such as multi-source HTL methods.

4. What is the performance of the method that does not use the group information to implement the domain alignment? I did not see the results of this kind of method.

5. In Problem 1, n_t is less than 7 seems too extreme?

6. Grammar issues should be addressed in the revised version.


**Time Spent Reviewing:**

12 hours

---

> ### Author Response · Authors · 2021-08-09
> **Response to Reviewer 4XQu**
>
> Thanks for your constructive comments! We will answer your major concerns in the following.
>
> >Q1. What is the key assumption that makes FHA solvable?
>
> A1: Like unsupervised domain adaptation (UDA), the key assumption that makes FHA solvable is that the combined risk  of source domain and target domain is small [r1, r2]. In terms of the DA learning bound, if the  is large, the target risk is also large, indicating a poor adaptation performance.
>
> >Q2. How could you know the initial steps being good? How could we ensure we can get good results from possible bad target samples in the loop mentioned above?
>
> A2: We have thought about this issue, and we find that the quality of intermediate data is really bad at the beginning of training procedure. Therefore, in Algorithm 1, we only train the generator before the $T_{max}-T_{f}$ epoch, and then we train the generator and target model simultaneously.
>
> >Q3. Network-based HTL methods should be also considered as a baseline. Besides, more HTL methods should be reviewed, such as multi-source HTL methods.
>
> A3: In Appendix F, we compare TOHAN with dkdHTL that is a novel HTL method [41]. In the revision, we will add more reviews about various HTL methods.
>
> >Q4. What is the performance of the method that does not use the group information to implement the domain alignment?
>
> A4: We have tried to implement domain alignment with class information, instead of group information. Specifically, we train the target classifier with target data and source model under the framework of adversarial domain adaptation. We find the experimental results are bad consistently. The reason is that the target data are too few. Therefore, in order to augment data, we generate intermediate data and pair them with target data to obtain four groups.
>
> >Q5. In Problem 1, $n_t$ is less than 7 seems too extreme?
>
> A5: Our paper mainly focuses on the few shot scenario. Following [r2], we choose $n_t$≤7 target data per class.
>
> >Q6. Grammar issues should be addressed in the revised version.
>
> A6: Thanks for your sincere reminder. We will check the grammar issues carefully and modify them in the revised version.
>
> References
>
> [r1] Ben-David, Shai, et al. "A theory of learning from different domains." Machine learning 79.1 (2010): 151-175.
>
> [r2] Zhong, Li, et al. "How does the Combined Risk Affect the Performance of Unsupervised Domain Adaptation Approaches?."  AAAI2021.

---

> > ### Comment · Reviewer_4XQu · 2021-08-17
> > **the rebuttal has addressed my concerns.**
> >
> > After reading the rebuttal, I feel satisfied and the rebuttal has addressed my concerns.

---

### Official Review · Reviewer_FhwF · 2021-07-16

**Rating:** 6
**Confidence:** 4

**Summary:**

This paper focuses on solving few-shot domain adaptation problem without accessing to source data during the training time, i.e., hypothesis transfer. The proposed method "TOHAN" uses two DNNs: one learns intermediate domain while another minimize the target risk and intermediate-target discrepancy.

**Limitations And Societal Impact:**

Generally, the problem setting is interesting and the solution is reasonable. My major concerns are:
1. How much time does it take running one epoch?
2. Since there are only a few target data, is there a balanced problem between target data and the intermediate data?
3. Since the distance between intermediate data and the target data is minimized in the following step, does it important to make the intermediate data look like target data in the generation step?

**Main Review:**

Hypothesis transfer that this paper focuses on is an interesting and important problem which can protect pravicy. Also, only get access to a few labeled target data makes this problem more challenging. By solving the few-shot problem, the proposed method generates intermediate data using a generator and fixed source model. Then the distribution discrepancy between intermediate data and the target data are minimized.

**Time Spent Reviewing:**

5

---

> ### Author Response · Authors · 2021-08-09
> **Response to Reviewer FhwF**
>
> Thanks for your constructive comments! We will answer your major concerns in the following.
>
> >Q1. How much time does it taking running one epoch?
>
> A1: For digital tasks (MNIST, SVHN, USPS), it takes about five minutes to run one epoch . While, for objective tasks (CIFAR-10, STL-10), it takes about nine minutes. Our experiments are conducted on one NVIDIA RTX 2080Ti GPU.
>
> >Q2. Since there are only a few target data, is there a balanced problem between target data and the intermediate data?
>
> A2: There is no the balanced problem. The training data of TOHAN are all intermediate-target data pair (i.e., the four groups demonstrated in our paper). Although there are only a few target data, it accounts for exactly the same proportion of the training samples as the intermediate data.
>
> >Q3. Since the distance between intermediate data and the target data is minimized in the following step, does it important to make the intermediate look like target data in the generation step.
>
> A3: Yes, our final goal is to generate data that is close to target data. It is worth noting that both minimizing the distance between intermediate data and the target data and making the intermediate data fit for source model are synchronous in the generation step. It can be seen in the Eq. (7).

---

> > ### Author Response · Authors · 2021-08-27
> > **We are glad to receive your valuable comments!**
> >
> > Dear Reviewer FhwF,
> >
> > It is glad to receive your valuable comments! We have responded to all of your comments in the last thread.
> >
> > Have our responses addressed your major concerns?
> >
> > We are glad to hear from you and happy to improve the quality of our paper according to your valuable comments!
> >
> > Best,
> >
> > Authors of Paper2423

---

### Decision · Program_Chairs · 2021-09-27

**Decision:**

Accept (Spotlight)

**Comment:**

The paper provides a few-shot learning technique that avoids looking at the data obtained to train previous tasks. As such it has promise for privacy preserving settings. The techniques seem solid, as well as the experimental setting. There where a few concerns related to over-promising in the area of differential privacy, however the reviewers agree that this can be fixed by the authors towards a camera-ready version. I recommend accepting the paper - it should be a good addition to Neurips